# EyeTrackUAV2: A Large-Scale Binocular Eye-Tracking Dataset for UAV Videos

**Anne-Flore Perrin** [1,*] , **Vassilios Krassanakis** [2,4] , **Lu Zhang** [3] , **Vincent Ricordel** [2] ,
**Matthieu Perreira Da Silva** [2] **and Olivier Le Meur** [1]

1   IRISA, CNRS, Univ Rennes, 263 Avenue Général Leclerc, 35000 Rennes, France; olemeur@irisa.fr
2   Laboratoire des Sciences du Numérique de Nantes (LS2N), Polytech Nantes, Université de Nantes,
    44306 Nantes CEDEX 3, France; Vincent.Ricordel@univ-nantes.fr (V.R.);
    matthieu.perreiradasilva@univ-nantes.fr (M.P.D.S.)
3   IETR, CNRS, INSA Rennes, IETR-UMR 6164, 35000 Rennes, France; lu.ge@insa-rennes.fr
4   Department of Surveying and Geoinformatics Engineering, University of West Attica,
    28 Agiou Spyridonos Str., 12243, Aigaleo, Greece ; krasvas@uniwa.gr
*   Correspondence: anne-flore.perrin@irisa.fr; Tel.: +33-299-842573

**Abstract:** The fast and tremendous evolution of the unmanned aerial vehicle (UAV) imagery gives place to the multiplication of applications in various fields such as military and civilian surveillance, delivery services, and wildlife monitoring. Combining UAV imagery with study of dynamic salience further extends the number of future applications. Indeed, considerations of visual attention open the door to new avenues in a number of scientific fields such as compression, retargeting, and decision-making tools. To conduct saliency studies, we identified the need for new large-scale eye-tracking datasets for visual salience in UAV content. Therefore, we address this need by introducing the dataset EyeTrackUAV2. It consists of the collection of precise binocular gaze information (1000 Hz) over 43 videos (RGB, 30 fps, 1280×720 or 720×480). Thirty participants observed stimuli under both free viewing and task conditions. Fixations and saccades were then computed with the dispersion-threshold identification (I-DT) algorithm, while gaze density maps were calculated by filtering eye positions with a Gaussian kernel. An analysis of collected gaze positions provides recommendations for visual salience ground-truth generation. It also sheds light upon variations of saliency biases in UAV videos when opposed to conventional content, especially regarding the center bias.

**Keywords:** dataset; salience; unmanned aerial vehicles (UAV); videos; visual attention; eye tracking; surveillance

## 1. Introduction

For a couple of decades now, we have witnessed the fast advances and growing use of UAV for multiple critical applications. UAV refers here to unmanned aerial vehicles, autonomous or monitored from remote sites. This imagery enables a broad range of uses, from making vacation movies to drone races for mainstream civilian applications. Tremendous professional services are developed, among others fire detection [1], wildlife counting [2], journalism [3], precision agriculture, and delivery services. But most applications are military, from aerial surveillance [4], drone-based warfare [5] to moving targets tracking [6], object, person, and anomaly detection [7–9].

The UAV imagery proposes a new representation of visual scenes that makes all these new applications possible. UAV vision is dominant and hegemonic [10]. The bird point of view modifies the perspective, size and features of objects [11]. It introduces a loss of pictorial depth cues [12] such as horizontal line [13]. Also, UAV high autonomy in conjunction with large-field of view camera permits

to cover large areas in limited time duration. Besides, embedded sensors can be multi-modal and can include RGB, thermal, IR, or multi-spectral sensors. Multiplying imagery modalities allows overcoming possible weaknesses of RGB-only cameras [10]. For instance, occlusions may be compensated by thermal information, and the capture of IR is desired for low-luminance environments [14].

UAV scene depiction is rich, comprehensive, and promising, which explains its success. However, challenges to come are even more compelling. Edney-Browne [10] wondered how the capacity of UAV capturing the external reality (visuality) is related to perceptual and cognitive vision in humans. Variations in UAV characteristics such as perspective view and object size may change viewers' behavior towards content. Consequently, new visual attention processes may be triggered for this specific imaging. This means that studying UAV imagery in light of human visual attention not only opens the door to plenty of applications but could also enable to gather further knowledge on perceptual vision and cognition.

In the context of UAV content, there are very few eye-tracking datasets. This is the reason why we propose and present in this paper a new large-scale eye-tracking dataset, freely downloadable from the internet. This dataset aims to strengthen our knowledge on human perception and could play a fundamental role for designing new computational models of visual attention.

The paper is organized as follows. In Section 2, we first justify and elaborate on the need for large-scale eye-tracking databases for UAV videos. Then, we introduce the entire process of dataset creation in Section 3. It describes the content selection, the experiment set up, and the implementation of fixations, saccades, and gaze density maps. Section 4 presents an in-depth analysis of the dataset. The study is two-fold: it explores what ground truth should be used for salience studies, and brings to light the fading of conventional visual biases UAV stimuli. Finally, conclusions are provided in Section 5.

## 2. Related Work

Visual attention occurs to filter and sort out visual clues. Indeed, it is impossible to process simultaneously all the information of our visual field. Particular consideration should be dedicated to identifying which attentional processes are involved as they are diverse and aim at specific behaviors. For instance, one must make the distinction between overt and covert attention [15]. The former refers to a direct focus where eyes and head point. The latter relates to the peripheral vision, where attention is directed without eye movement towards it. In practice, when an object of interest is detected in the area covered by the covert attention, one may make a saccade movement to direct the eyes from the overt area to this position. The context of visualization is also important. For instance, we make a distinction between two content exploration processes [16]: (1) A no constraint examination named free viewing. The observer is rather free from cognitive loads and is supposed to mainly use bottom-up or exogenous attention processes driven by external factors, e.g., content and environment stimuli. (2) A task-based visualization, such as surveillance for instance. Cognitive processes such as prior knowledge, wilful plans, and current goals guide the viewer's attention. This is known as top-down or endogenous attention. A strict division is slightly inaccurate in that both top-down and bottom-up processes are triggered during a visual stimuli in a very intricate interaction [17,18].

In computer science, it is common to study bottom-up and top-down processes through the visual salience. Visual salience is a representation of visual attention in multimedia content as a probability distribution per pixels [19]. Salience analyses rest on the relation of visual attention to eye movements, and these latter are obtained through gaze collection with eye-trackers [20]. Saliency predictions help to understand computational cognitive neuroscience as it reveals attentional behaviors and systematic viewing tendencies such as center bias [17]. Multiple applications derive from saliency predictions such as compression [21], content-aware re-targeting, object segmentation [22], and detection [23,24].

Recently, there has been a growing interest on one particular application, which combines visual salience and UAV content. Information overload in the drone program and fatigue in military operators may have disastrous consequences for military applications [10]. New methods and approaches are

required to detect anomaly in UAV footages and to ease the decision-making. Among them, we believe that computational models of visual attention could be used to simulate operators' behaviors [25]. Eventually, thanks to predictions, operators' workloads can be reduced by eliminating unnecessary footages segments. Other works support the use of salience to enhance the efficiency of target-detection task completion. For instance Brunyé et al. [26] studied the combination of salience (in terms of opacity with the environment) and biological motion (presence and speed) in textured backgrounds. They concluded that salience is very important for slowly moving objects, such as camouflaged entities. Meanwhile, fast biological movements are highly attention-grabbing, which diminishes the impact of static salience. Accordingly, it makes sense to develop dynamic saliency models tailored to UAV content.

However, we demonstrate in [27] that current saliency models lack efficiency in terms of prediction for UAV content. This applies to all types of prediction models, handcrafted features and architecture implementing deep learning to a lesser extent, whether they are static or dynamic schemes. Typical handcrafted and low-level features learnt on conventional imaging may not suit UAV content. Besides, in conventional imaging the center position is the best location to have access to most visual information of a content [28]. This fact leads to a well-known bias in visual attention named central bias. This effect may be associated with various causes. For instance, Tseng et al. [29] showed a contribution of photographer bias, viewing strategy, and to a lesser extent, motor, re-centering, and screen center biases to the center bias. They are briefly described below:

- The photographer bias often emphasizes objects in the content center through composition and artistic intent [29].
- Directly related to photographer bias, observers tend to learn the probability of finding salient objects at the content center. We refer to this behavior as a viewing strategy.
- With regards to the human visual system (HVS), the central orbital position, that is when looking straight ahead, is the most comfortable eye position [30], leading to a recentering bias.
- Additionally, there is a motor bias, in which one prefers making short saccades and horizontal displacements [31,32].
- Lastly, onscreen presentation of visual content pushes observers to stare at the center of the screen frame [28,33]. This experimental bias is named the screen center bias.

The central bias is so critical in the computational modelling of visual attention that saliency models include this bias as prior knowledge or use it as a baseline to which saliency predictions are being compared [34]. The center bias is often represented by a centered isotropic Gaussian stretched to the video frame aspect ratio [35,36]. The presence of this bias in UAV videos has already been questioned in our previous work [27]. We showed that saliency models that heavily rely on the center bias were less efficient on Unmanned Aerial Vehicle (UAV) videos than on conventional video sequences. Therefore, we believe that the central bias could be less significant in drone footage as a result of the lack of photographer bias or due to UAV content characteristics. It would be beneficial to evaluate qualitatively and quantitatively the center bias on a larger dataset of UAV videos to support our assumption.

While it is now rather easy to find eye tracking data on typical images [35,37–45] or videos [46–50], and that there are many UAV content datasets [7,51–62], it turns out to be extremely difficult to find eye-tracking data on UAV content. This is even truer when we consider dynamic salience, which refers to salience for video content. To the best of our knowledge, EyeTrackUAV1 dataset released in 2018 [11] is the only public dataset available for studying the visual deployment over UAV video. There exists another dataset AVS1K [63]. However, AVS1K is, to the present day, not publicly available. We thus focus here on EyeTrackUAV1, with the awareness that all points below but the last apply to AVS1K.

EyeTrackUAV1 consists of 19 sequences (1280×720 and 30 fps) extracted from the UAV123 database [55]. The sequence selection relied on content characteristics, which are the diversity of environment, distance and angle to the scene, size of the principal object, and the presence of sky.

Precise binocular gaze data (1000 Hz) of 14 observers were recorded under free viewing condition, for every content. Overall, the dataset comprises eye-tracking information on 26,599 frames, which represents 887 seconds of video. In spite of a number of merits, this dataset presents several limitations for saliency prediction applications. These limitations have been listed in [27]. We briefly summarize them below:

- UAV may embed multi-modal sensors during the capture of scenes. Besides conventional RGB cameras, to name but a few thermal, multi-spectral, and infrared cameras consist of typical UAV sensors. Unfortunately, EyeTrackUAV1 lacks non-natural content, which is of great interest for the dynamic field of salience. As already mentioned, combining content from various imagery in datasets is advantageous for numerous reasons. It is necessary to continue efforts toward the inclusion of more non-natural content in databases.
- In general, the inclusion of more participants in the collection of human gaze is encouraged. Indeed, reducing variable errors by including more participants in the eye tracking experiment is beneficial. It is especially true in the case of videos as salience is sparse due to the short displaying duration of a single frame. With regards to evaluation analyses, some metrics measuring similarity between saliency maps consider fixation locations for saliency comparison (e.g., any variant of area under the curve (AUC), normalized scanpath saliency (NSS), and information gain (IG)). Having more fixation points is more convenient for the use of such metrics.
- EyeTrackUAV1 contains eye-tracking information recorded during free-viewing sessions. That is, no specific task was assigned to observers. Several applications for UAV and conventional imaging could benefit from the analysis and reproduction of more top-down attention, related to a task at hand. More specifically, for UAV content, there is a need for specialized computational models for person or anomaly detection.
- Even though there are about 26,599 frames in EyeTrackUAV, they come from only 19 videos. Consequently, this dataset just represents a snapshot of the reality. We aim to go further by introducing more UAV content.

To extend and complete the previous dataset and to tackle these limitations, we have created the EyeTrackUAV2 dataset, introduced below.

## 3. EyeTrackUAV2 Dataset

This section introduces the new dataset EyeTrackUAV2 aiming at tackling issues mentioned above. EyeTrackUAV2 includes more video content than its predecessor EyeTrackUAV1. It involves more participants, and considers both free and task-based viewing. In the following subsections, we first elaborate on the selection of video content, followed by a description of the eye-tracking experiment. It includes the presentation of the eye-tracking apparatus, the experiment procedure and setup, and the characterization of population samples. Finally, we describe the generation of the human ground truth, i.e., algorithms for fixation and saccade detection as well as gaze density map computation.

### 3.1. Content Selection

Before collecting eye-tracking information, experimental stimuli were selected from multiple UAV video datasets. We paid specific attention to select videos suitable for both free and task-based viewing as experimental conditions. Also, the set of selected videos had to cover multiple UAV flight altitudes, main surrounding environments, main sizes of observed objects and angles between the aerial vehicle and the scene, as well as the presence or not of sky. We consider these characteristics favor the construction of a representative dataset of typical UAV videos, as suggested in [11].

We examined the following UAV datasets: UCF's dataset (http://crcv.ucf.edu/data/UCF_Aerial_Action.php), VIRAT [51], MRP [52], the privacy-based mini-drones dataset [53], the aerial videos dataset described in [54], UAV123 [55], DTB70 [57], Okutama-Action [58], VisDrone [64], CARPK [59], SEAGULL [60], DroneFace [61], and the aerial video dataset described in [56]). A total of 43 videos

(RGB, 30 frame per second (fps), 1280×720 or 720×480) were selected from databases VIRAT, UAV123, and DTB70. These three databases exhibited different contents for various applications, which made the final selection representative of the UAV ecosystem. We present below the main characteristics of the three selected datasets:

- UAV123 included challenging UAV content annotated for object tracking. We restricted the content selection to the first set, which included 103 sequences (1280×720 and 30 fps) captured by an off-the-shelf professional-grade UAV (DJI S1000) tracking various objects in a range of altitudes comprised between 5–25 m. Sequences included a large variety of environments (e.g., urban landscapes, roads, and marina), objects (e.g., cars, boats, and persons) and activities (e.g., walking, biking, and swimming) as well as presenting many challenges for object tracking (e.g., long- and short-term occlusions, illumination variations, viewpoint change, background clutter, and camera motion).
- Aerial videos in the VIRAT dataset were manually selected (for smooth camera motion and good weather conditions) from rushes of a total amount of 4 h in outdoor areas with broad coverage of realistic scenarios for real-world surveillance. Content included "single person", "person and vehicle", and "person and facility" events, with changes in viewpoints, illumination, and visibility. The dataset came with annotations of moving object tracks and event examples in sequences. The main advantage of VIRAT videos was its perfect fit for military applications. It covered fundamental environment contexts (events), conditions (rather poor quality and weather condition impairments), and imagery (RGB and IR). We decided to keep the original resolution of videos (720×480) to prevent the introduction of unrelated artifacts.
- The 70 videos (RGB, 1280×720 and 30 fps) from DTB70 dataset were manually annotated with bounding boxes for tracked objects. Sequences were shot with a DJI Phantom 2 Vision+ drone or were collected from YouTube to add diversity in environments and target types (mostly humans, animals, and rigid objects). There was also a variety of camera movements (both translation and rotation), short- and long-term occlusions, and target deformability.

Table 1 reports for each database the number of sequences selected, their native resolution, duration and frame number. Table 2 presents basic statistics of the database in terms of number of frames and duration.

**Table 1.** Stimuli original datasets.

| Dataset | Native Resolution | Proportion of Content Seen Per Degree of Visual Angle (%) | Videos Number | Frames Number (30 fps) | Duration (s) |
|---|---|---|---|---|---|
| VIRAT [51] | 720×480 | 1.19 | 12 | 17,851 | 595.03 |
| UAV123 [55] | 1280×720 | 0.44 | 22 | 20,758 | 691.93 |
| DTB70 [57] | 1280×720 | 0.44 | 9 | 3,632 | 121.07 |
| Overall | | | **43** | **42,241** | **1,408.03 (23:28 min)** |

**Table 2.** Basic statistics on selected videos.

| | Number of Frames | | | | Duration (MM:SS) | | | |
|---|---|---|---|---|---|---|---|---|
| | VIRAT | UAV123 | DTB70 | Overall | VIRAT | UAV123 | DTB70 | Overall |
| Total | 17,851 | 20,758 | 3,632 | 42,241 | 09:55 | 11:32 | 02:01 | 23:28 |
| Average | 1,488 | 944 | 404 | 982 | 00:50 | 00:31 | 00:13 | 00:33 |
| Standard Deviation | 847 | 615 | 177 | 727 | 00:28 | 00:21 | 00:06 | 00:24 |
| Minimum | 120 | 199 | 218 | 120 | 00:04 | 00:07 | 00:07 | 00:04 |
| Maximum | 3,178 | 2,629 | 626 | 3,178 | 01:46 | 01:28 | 00:21 | 01:46 |

*3.2. Content Diversity*

Figure 1 presents the diversity of selected UAV sequences by illustrating the first frame of every content. Visual stimuli cover a variety of visual scenes in different environments (e.g., public and military environments, roads, buildings, sports, port areas, etc.) and different moving or fixed objects (e.g., people, groups of people, cars, boats, bikes, motorbikes, etc.). Selected videos were captured from various flight heights and different angles between the UAV and the ground (allowing or not the presence of sky during their observation). Also, three sequences, extracted from the VIRAT dataset, were captured by IR cameras. Additionally, we considered various video duration as the length of the video may possibly impact the behavior of observers due to fatigue, resulting in a lack of attention and more blinking artifacts [10,65].

To quantitatively show the diversity of selected videos, we have computed temporal and spatial complexity [66], named temporal index (TI) ($\in [0, +\infty[$) and spatial index (SI) ($\in [0, +\infty[$), respectively. These features are commonly used in image quality domain for describing the properties of selected images. They characterize the maximum standard deviation of spatial and temporal discrepancies over the entire sequence. The higher a measure is, the more complex the content. TI and SI are reported per sequence in Table 3. The range of temporal complexity in sequences is broad, displaying the variety of movements present in sequences. Spatial measures are more homogeneous. Indeed, the spatial complexity is due to the bird point of view of the sensor. The aircraft high up position offers access to a large amount of information. Table 3 reports a number of information for all selected sequences.

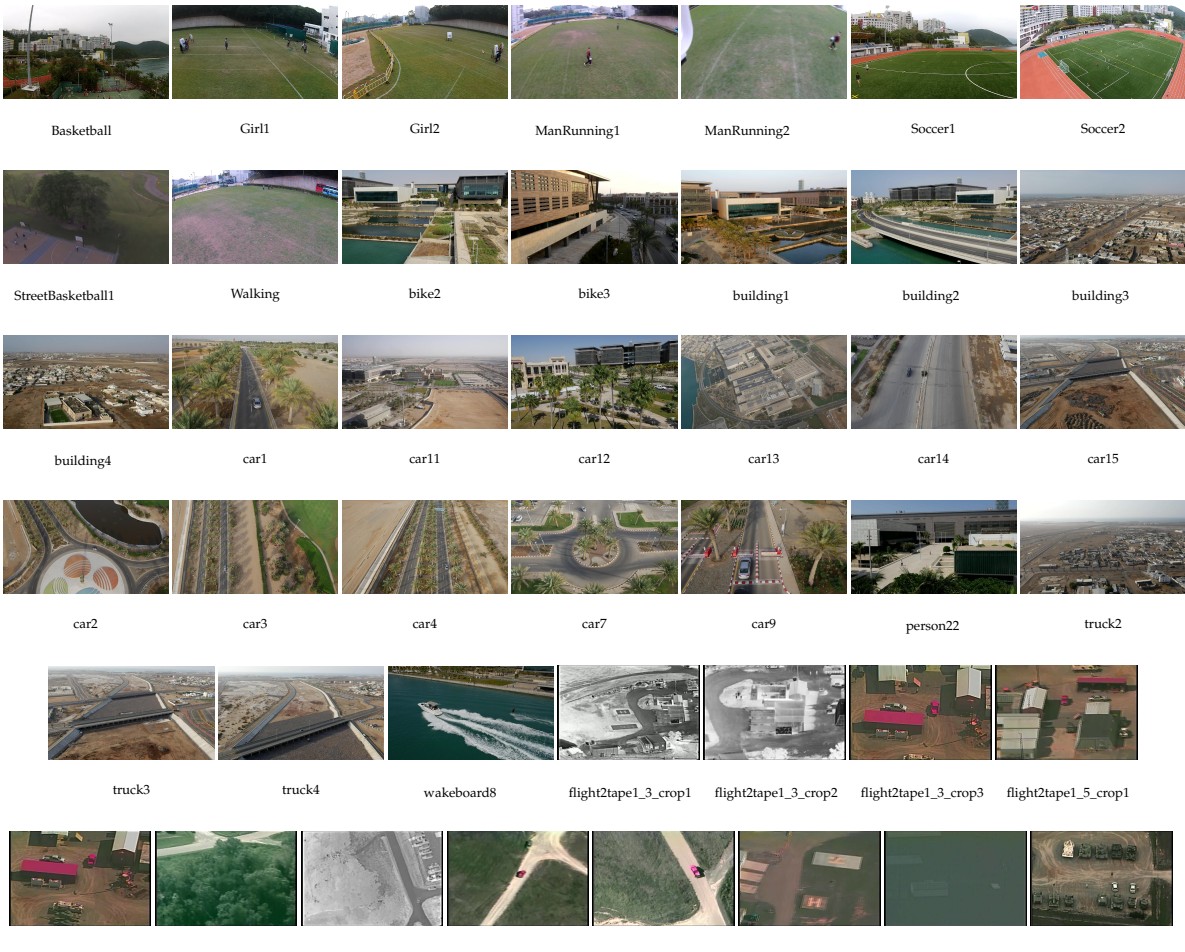

**Figure 1.** EyeTrackUAV2 dataset: first frame of each sequence.

**Table 3.** Stimuli ID and name, their original dataset, number of frames together with starting and ending frame number, duration, and native resolution.

| ID | Video | Dataset | Number of Frames | Start Frame | End Frame | Duration (ms) | SI | TI | Altitude | Environment | Object Size | Horizontal Line (Sea, Sky) | Main Angle |
|---|---|---|---|---|---|---|---|---|---|---|---|---|---|
| 1 | 09152008flight2tape1_3 (crop 1) | | 120 | 1 | 120 | 4,000 | 0.455 | 32 | High | Urban military - **IR** | Small | False | Oblique |
| 2 | 09152008flight2tape1_3 (crop 2) | | 367 | 137 | 503 | 12,234 | 0.474 | 35 | High | Urban military - **IR** | Small | False | Oblique |
| 3 | 09152008flight2tape1_3 (crop 3) | | 3,178 | 4,735 | 7,912 | 105,934 | 0.452 | 43 | Intermediate | Urban military | Medium, Small | False | Oblique |
| 4 | 09152008flight2tape1_5 (crop 1) | | 972 | 218 | 1,189 | 32,400 | 0.467 | 37 | Intermediate | Urban military | Medium, Small | False | Oblique |
| 5 | 09152008flight2tape1_5 (crop 2) | | 1,715 | 4,555 | 6,269 | 57,167 | 0.461 | 45 | Intermediate | Urban military | Medium, Small | False | Oblique |
| 6 | 09152008flight2tape2_1 (crop 1) | VIRAT | 1,321 | 1 | 1,321 | 44,034 | 0.484 | 40 | Intermediate, Low | Urban military | Medium, Big | False | Oblique |
| 7 | 09152008flight2tape2_1 (crop 2) | | 1,754 | 2,587 | 4,340 | 58,467 | 0.484 | 41 | High | Roads rural - **IR** | Small | False | Oblique |
| 8 | 09152008flight2tape2_1 (crop 3) | | 951 | 4,366 | 5,316 | 31,700 | 0.482 | 33 | Intermediate | Urban military | Medium, Big | False | Oblique |
| 9 | 09152008flight2tape2_1 (crop 4) | | 1,671 | 6,482 | 8,152 | 55,700 | 0.452 | 32 | High | Roads rural | Medium | False | Oblique, Vertical |
| 10 | 09152008flight2tape3_3 (crop 1) | | 2,492 | 3,067 | 5,558 | 83,067 | 0.474 | 42 | Intermediate | Urban military | Small | False | Oblique |
| 11 | 09162008flight1tape1_1 (crop 1) | | 1,894 | 1,097 | 2,990 | 63,134 | 0.448 | 39 | Low | Urban military, Roads rural | Medium, Small | False | Oblique |
| 12 | 09162008flight1tape1_1 (crop 2) | | 1,416 | 4,306 | 5,721 | 47,200 | 0.477 | 29 | Intermediate, High | Urban military | Small | False | Oblique |
| | Average | | 1,488 | | | 50,000 | 0.468 | 37.33 | | | | | |
| | Standard deviation | | 847 | | | 28,000 | 0.01 | 5.10 | | | | | |
| 13 | bike2 | | 553 | 1 | 553 | 18,434 | 0.468 | 22 | Intermediate | Urban, building | Small, Very small | True | Horizontal |
| 14 | bike3 | | 433 | 1 | 433 | 14,434 | 0.462 | 19 | Intermediate | Urban, building | Small | True | Horizontal |
| 15 | building1 | | 469 | 1 | 469 | 15,634 | 0.454 | 12 | Intermediate | Urban, building | Very Small | True | Horizontal |
| 16 | building2 | | 577 | 1 | 577 | 19,234 | 0.471 | 37 | Intermediate | Urban, building | Medium, Small | True | Horizontal |
| 17 | building3 | | 829 | 1 | 829 | 27,634 | 0.451 | 27 | High | Urban in desert | Small | True | Horizontal |
| 18 | building4 | | 787 | 1 | 787 | 26,234 | 0.464 | 29 | High, Intermediate | Urban in desert | None | True, False | Horizontal, Oblique |
| 19 | car1 | | 2,629 | 1 | 2,629 | 87,634 | 0.471 | 59 | Low, Intermediate | Road rural | Big, Medium | True | Oblique |
| 20 | car11 | | 337 | 1 | 337 | 11,234 | 0.467 | 31 | High | Suburban | Small | True, False | Horizontal, Oblique |
| 21 | car12 | | 499 | 1 | 499 | 16,634 | 0.467 | 39 | Low | Road urban, sea | Medium, Small | True | Horizontal |
| 22 | car13 | | 415 | 1 | 415 | 13,834 | 0.461 | 26 | High | Urban | Very very small | False | Oblique, Vertical |
| 23 | car14 | UAV123 | 1,327 | 1 | 1,327 | 44,234 | 0.471 | 25 | Low | Road suburban | Medium | False | Oblique |
| 24 | car15 | | 469 | 1 | 469 | 15,634 | 0.471 | 18 | Intermediate | Road towards urban | Small, Very small | True | Oblique |
| 25 | car2 | | 1,321 | 1 | 1,321 | 44,034 | 0.464 | 24 | Intermediate | Road rural | Medium | False | Oblique, Vertical |
| 26 | car3 | | 1,717 | 1 | 1,717 | 57,234 | 0.467 | 27 | Intermediate | Road rural | Medium | False | Oblique, Vertical |
| 27 | car4 | | 1,345 | 1 | 1,345 | 44,834 | 0.462 | 23 | Intermediate, Low | Road rural | Big | False | Oblique, Vertical |
| 28 | car7 | | 1,033 | 1 | 1,033 | 34,434 | 0.464 | 18 | Intermediate | Road suburban | Medium | False | Oblique |
| 29 | car9 | | 1,879 | 1 | 1,879 | 62,634 | 0.470 | 23 | Intermediate, Low | Road suburban | Medium | False, True | Oblique, Horizontal |
| 30 | person22 | | 199 | 1 | 199 | 6,634 | 0.456 | 31 | Low | Urban sea | Medium, Big | True | Horizontal |
| 31 | truck2 | | 601 | 1 | 601 | 20,034 | 0.453 | 24 | High | Urban road | Small | True | Horizontal |
| 32 | truck3 | | 535 | 1 | 535 | 17,834 | 0.472 | 18 | Intermediate | Road towards urban | Small, Very small | True | Oblique |
| 33 | truck4 | | 1,261 | 1 | 1,261 | 42,034 | 0.466 | 17 | Intermediate | Road towards urban | Small | True | Oblique, Horizontal |
| 34 | wakeboard8 | | 1,543 | 1 | 1,543 | 51,434 | 0.472 | 39 | Low | Sea urban | Medium, Big | True, False | Oblique, Vertical, Horizontal |
| | Average | | 944 | | | 31,000 | 0.465 | 26.73 | | | | | |
| | Standard deviation | | 615 | | | 21,000 | 0.01 | 10.14 | | | | | |
| 35 | Basketball | | 427 | 1 | 427 | 14,234 | 0.477 | 48 | Intermediate | Field suburban | Medium | True | Oblique |
| 36 | Girl1 | | 218 | 1 | 218 | 7,267 | 0.481 | 31 | Low | Field suburban | Big | True | Horizontal |
| 37 | Girl2 | | 626 | 1 | 626 | 20,867 | 0.482 | 30 | Low | Field suburban | Big | True | Horizontal |
| 38 | ManRunning1 | | 619 | 1 | 619 | 20,634 | 0.483 | 23 | Low | Field suburban | Big | True | Horizontal, Oblique |
| 39 | ManRunning2 | DTB70 | 260 | 1 | 260 | 8,667 | 0.484 | 27 | Low | Field suburban | Very big | False | Vertical, Oblique |
| 40 | Soccer1 | | 613 | 1 | 613 | 20,434 | 0.476 | 57 | Low, Intermediate | Field suburban | Very big, Big | True | Horizontal |
| 41 | Soccer2 | | 233 | 1 | 233 | 7,767 | 0.475 | 24 | High | Field suburban | Small | True | Oblique |
| 42 | StreetBasketball1 | | 241 | 1 | 241 | 8,034 | 0.379 | 37 | Low | Field urban | Big | True, False | Oblique, Vertical |
| 43 | Walking | | 395 | 1 | 395 | 13,167 | 0.476 | 31 | Low | Field suburban | Big, Very big | True | Oblique |
| | Average | | 404 | | | 13,000 | 0.468 | 34.22 | | | | | |
| | Standard deviation | | 177 | | | 6,000 | 0.03 | 11.39 | | | | | |
| | Average | | 982 | | | 33 s | 0.466 | 31.26 | | | | | |
| | Standard deviation | | 727 | | | 24 s | 0.02 | 10.30 | | | | | |
| | Overall | | 42,241 | | | 1,408 s | | | | | | | |

*3.3. Experimental Design*

To record the gaze deployment of subjects while viewing UAV video sequences displayed onscreen, it was required to define an experimental methodology. All the details are presented below.

3.3.1. Eye-Tracking Apparatus

A specific setup was designed to capture eye-tracking information on video stimuli. It included a rendering monitor, an eye-tracking system, a control operating system, and a controlled laboratory test room. Figure 2 illustrates the experimental setup used during the collection of gaze information. We can observe the arrangement of all the systems described below.

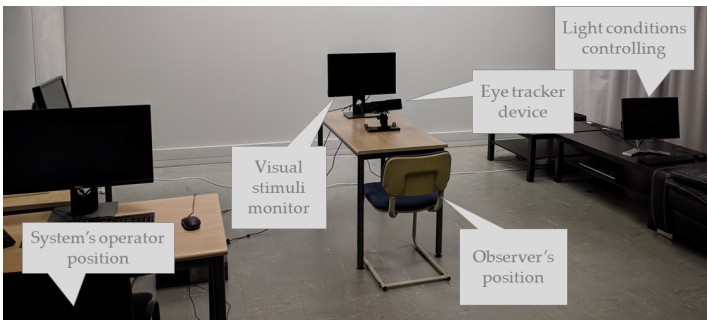

**Figure 2.** Experiment setup.

To run the experiment and collect gaze information, we used the EyeLink® 1000 Plus eye-tracking system (https://www.sr-research.com/eyelink-1000-plus/), in the head free-to-move *remote mode*, taking advantage of its embedded 25 mm camera lens. The eye tracker principle is to detect and record the IR illuminator reflection rays on the observer's pupil [65]. This system enables the collection of highly precise gaze data at a temporal frequency of 1000 Hz and a spatial accuracy between the visual angle range of 0.25 and 0.50 degree, according to the manufacturer. The eye tracker's camera was configured for each subject, without affecting the corresponding distance between them. This configuration guarantees to achieve an optimal detection of the observer's eyes and head sticker.

The experimental monitor which displayed stimuli was a 23.8-inch (52.70×29.65 cm) DELL P2417H computer monitor display (https://www.dell.com/cd/business/p/dell-p2417h-monitor/pd) with full HD resolution (1920×1080) at 60 Hz and with a response time of 6 ms. As suggested by both the International Telecommunication Union (ITU)-Broadcasting Service (television) (BT).710 [67] and manufacturer, observers sat in distances of about 3 H (1m ± 10cm) from the monitor, where H corresponds to the stimuli display height so that observers had an assumed spatial visual angle acuity of one degree. Moreover, the eye tracker camera was placed 43 cm away from the experimental display, and thus about 67 cm from participants. Based on this setting, there were 64 pixels per degree of visual angle in each dimension, and the display resolution was about 30×17 visual degrees.

Regarding software, the MPC-HC video player (https://mpc-hc.org/), considered as one of the most lightweight open-source video players, rendered the experimental video stimuli. Also, we took advantage of the Eyelink toolbox [68] as it is part of the third version of Psychophysics Toolbox Psychtoolbox-3 (PTB-3) (http://psychtoolbox.org/) and added in-house communication processes (LS2N, University of Nantes) for sync between control and display systems. The control system consists of an additional computer, used by the experimenter to configure and control the eye-tracking system with an Ethernet connection.

Eventually, eye-tracking tests were performed in a room with controlled constant light conditions. The performed calibration set the constant ambient light conditions at approximately 36.5 cd/m$^2$, i.e., 15% of the maximum stimuli monitor brightness—249 cm/m$^2$—as recommended by the ITU-BT.500 [69], with the i1 Display Pro X-Rite® system.

### 3.3.2. Stimuli Presentation

The random presentation of stimuli in their native resolution centered on the screen prevents ordering, resizing, and locating biases. Knowing that the monitor resolution is higher than that of selected sequences, video stimuli were padded with mid-grey. Additionally, to avoid possible biases in gaze allocation, a 2-second sequence of mid-gray frames was presented before playing a test sequence. Please note that the amount of original information contained in a degree of visual angle was not the same for VIRAT sequences than for other database content, as specified in Table 1.

Before starting the experiment, a training session was organized to get the subject familiar with the experiment design. It included a calibration procedure and its validation followed by the visualization of one video. This UAV video was the sequence car4 from the DTB70 dataset. To avoid any memory bias, this sequence was not included in test stimuli. Once subjects completed the training session, they could ask questions to experimenters before taking part into test sessions.

Regarding test sessions, they started with calibration and its validation. Then followed the visualization of nine videos during which subjects did or did not perform a task. To ensure the optimal quality of the collected gaze data, each participant took part in five test sessions. Splitting the experiment into sessions decreased the tiredness and lack of attention in observers. Also, this design enabled frequent calibration so that recordings did not suffer from the decrease of accuracy in gaze recordings with time [65].

With regards to calibration, the eye-tracking system was calibrated for each participant, following a typical 13 fixed-point detection procedure [65]. Actually, experimenters started tests with a nine-point strategy for calibration (subject 1 to 17 in free viewing (FV)) but realized that using a 13-point calibration produced more accurate gaze collection. The calibration reached validation when the overall deviation of both eye positions was approximately below the fovea vision accuracy (e.g., a degree of visual angle [65,70]). The calibration procedure was repeated until validation.

The participation of an observer in the experiment lasted about 50 minutes. It included test explanations, forms signing, and taking part in the training and the five test sessions. This duration was acceptable regarding the number of sessions and the fatigue in subjects.

### 3.3.3. Visual Tasks to Perform

EyeTrackUAV2 aimed to investigate two visual tasks. Indeed, we wanted to be able to witness visual attention processes triggered by top-down (or goal-directed) and bottom-up (or stimulus-driven) attention. Accordingly, we defined two visual tasks participants had to perform: the first condition was an FV task while the second relates to a surveillance viewing task (Task). The former task is rather common in eye-tracking tests [32,37,43,45,50,71]. Observers were simply asked to observe visual video stimuli without performing any task. For the surveillance-viewing task, participants were required to watch video stimuli and to push a specific button on a keyboard each time they observe a new—meaning not presented before—moving object (e.g., people, vehicle, bike, etc.) in the video. The purpose of this task was to simulate one of the basic surveillance procedures in which targets could be located anywhere when the visual search process was performed [72]. After reviewing typical surveillance systems' abilities [73], we decided to define our task as object detection. The defined object detection task was compelling in that it encompassed target-specific training (repeated discrimination of targets and non-targets) and visual search scanning (targets potentially located anywhere) [72]. The surveillance-viewing task is especially interesting for a military context, in which operators have to detect anomaly in drone videos.

### 3.3.4. Population

Overall, 30 observers participated in each phase of the test. Tested population samples were different for these two viewing conditions. They were carefully selected to be as diverse as possible. For instance, they include people from more than 12 different countries, namely Algeria (3%), Brazil,

Burundi, China, Colombia (10%), France (67%), Gabon, Guinea, South Arabia, Spain, Tunisia, and Ukraine. Additionally, we achieved gender and almost eye-dominance balance in both phases tests. Table 4 presents the detailed population characteristics for both tasks.

Each observer had been tested for visual acuity and color vision with Ishihara and Snellen tests [74,75]. Any failure of these tests motivated the dismissal of the person from the experiment. Before running the test, the experimenter provided subjects with written consent and information forms, together with oral instructions. This process made sure of the consent of participants and their understanding of the experiment process. It also ensured anonymous data collection.

**Table 4.** Population characteristics.

| Sample Statistics | FV | Task | Total |
|---|---|---|---|
| Participants | 30 | 30 | 60 |
| Female | 16 | 16 | 32 |
| Male | 14 | 14 | 28 |
| Average age | 31.7 | 27.9 | 29.8 |
| Std age | 11.0 | 8.5 | 10.0 |
| Min age | 20 | 19 | 19 |
| Max age | 59 | 55 | 59 |
| Left dominant eye | 19 | 9 | 28 |
| Right dominant eye | 11 | 21 | 32 |
| Participants with glasses | 0 | 4 | 4 |

*3.4. Post-Processing of Eye-Tracking Data*

First, we transformed collected raw signals into the pixel coordinate system of the original sequence. This conversion led to what we refer to as binocular gaze data. Let us be precise that the origin of coordinates was the top-left corner. Then, any gaze coordinates out of range were evicted, as they did not represent visual attention on stimuli. Once transformed and filtered, we extracted fixation and saccade information and created gaze density maps from gaze data. The remainder of this section describes all post-processing functions.

3.4.1. Raw Data

At first, coordinates of the collected binocular gaze data were transformed into the pixel coordinate system of the visual stimulus. Additionally, we addressed the original resolution of sequences. Coordinates outside the boundaries of the original resolution of the stimulus were filtered out as they were not located in the video stimuli display area. The following formula presents how the collected coordinates are transformed for both eyes:

$$
\begin{cases}
x_S &= \lfloor x_D - \frac{R_D^X - R_S^X}{2} \rfloor \\
y_S &= \lfloor y_D - \frac{R_D^Y - R_S^Y}{2} \rfloor
\end{cases}
\tag{1}
$$

where, $(x_S, y_S)$ and $(x_D, y_D)$ are the spatial coordinates on the stimulus and on the display, respectively. The operator $\lfloor . \rfloor$ allowed us to keep the coordinates if the coordinates were within the frame of the stimulus. Otherwise, the coordinate was discarded. $(R_S^X, R_S^Y)$ and $(R_D^X, R_D^Y)$ represent the stimulus resolution and the display resolution, respectively. For more clarity, Figure 3 displays the terms of the equation. Once this remapping had been done for both eyes, the spatial binocular coordinates were simply given by the average of the spatial coordinates of left and right eyes.

During the surveillance-viewing task, subjects pushed a button when detecting an object (never seen before) in the content. Triggering this button action was included in raw data. Consequently, we denotde in raw data a button activation (respectively no detection reaction) with the Boolean value 1

(respectively 0). Besides, for convenience, we extracted the positions of the observer's dominant eyes and included them in raw gaze data.

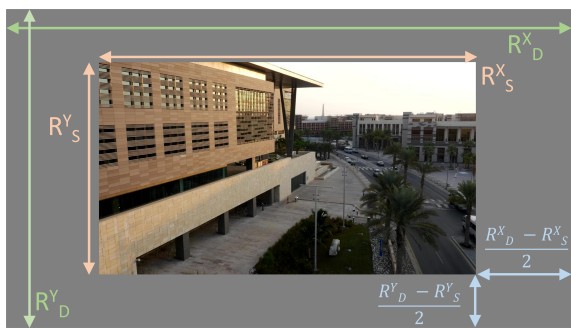

**Figure 3.** Stimulus displayed in its native resolution, and padded with mid-gray to be centered. Colored information relates to Equation (1).

### 3.4.2. Fixation and Saccade Event Detection

To retrieve fixations from eye positions, we used the dispersion-threshold identification (I-DT) [76] from the EyeMMV and LandRate toolboxes [77,78]. This algorithm performed "two-step" spatial and temporal thresholds. As exposed in [78,79], thanks to the very high precision of our eye-tracking equipment, we could combine the two-step spatial thresholds into one operation, as both thresholds had the same value. Ultimately, in our context, this algorithm conceptually implemented a spatial noise removal filter and a temporal threshold indicating the minimum fixation duration. We have selected the minimum threshold values from the state of the art to ensure the performance of the fixation detection algorithm. Accordingly, spatial and temporal thresholds were selected to be equal to 0.7 degree of visual angle and 80 ms [80], respectively. Finally, saccade events were calculated based on the computed fixations considering that a saccade corresponds to eye movements between two successive fixation points. When considering raw data of the dominant eye, I-DT exhibited a total number of fixations of 1,239,157 in FV and 1,269,433 in Task.

### 3.4.3. Human Saliency Maps

Saliency maps are a 2D topographic representation indicating the ability of an area to attract observers' attention. It is common to represent the salience of an image thanks to either its saliency map or by its colored representation, called heatmap. Human saliency maps are usually computed by convolving the fixation map, gathering observers' fixations, with a Gaussian kernel representing the foveal part of our retina. More details can be found in [71]. In our context, we did not compute convolved fixation maps. We took benefit from the high frequency of acquisition of the eye-tracker system to compute human saliency maps directly from raw gaze data (in pixel coordinates). For the sake of clarity, we from now will refer to the generated saliency maps as gaze density maps.

To represent salient regions of each frame, we followed the method described in [77]. We derived parameters from the experimental setup (e.g., a grid size of a pixel, a standard deviation of 0.5 degree of angle i.e., $\sigma = 32$ pixels, and a kernel size of $6\sigma$). For visualization purposes, gaze density maps were normalized between 0 and 255. Figure 4 presents gaze density maps obtained for both attention conditions in frame 100 of seven sequences. We selected frame 100 to get free from the initial center-bias in video exploration occurring during the first seconds. These examples illustrate the sparsity of salience in videos in free viewing, while task-based attention usually presents more salient points, more dispersed in the content than FV, depending on the task and attention-grabbing objects.

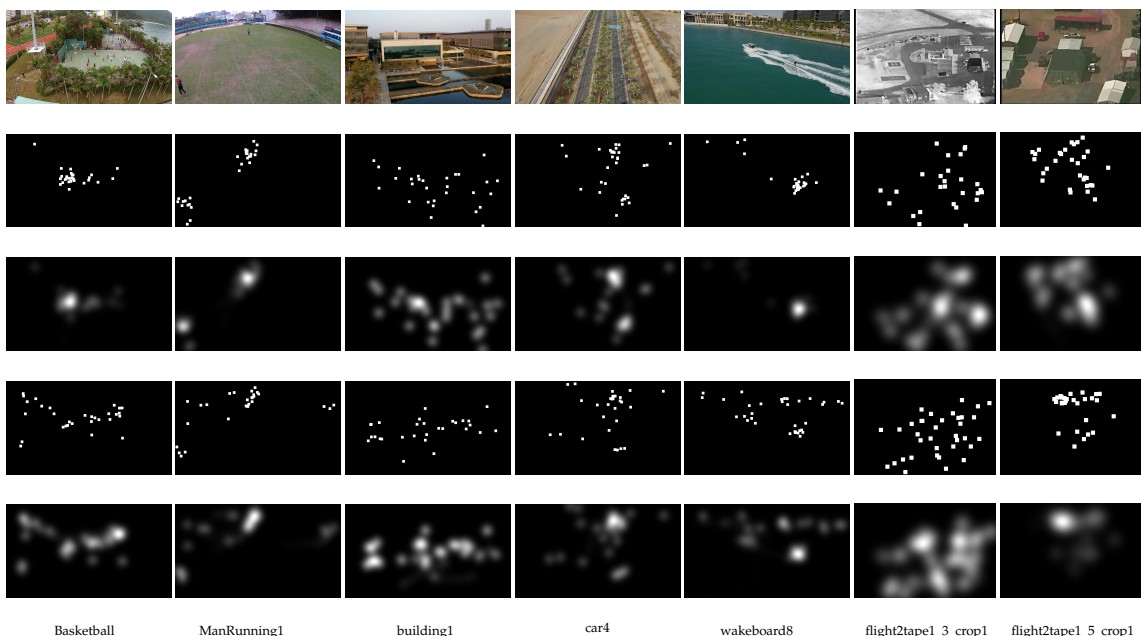

**Figure 4.** Frame 100 of seven sequences of EyeTrackUAV2 dataset, together with gaze density and fixation maps generated based on gaze data of dominant eye. Results are presented for both types of attention. The first row presents sequences hundredth frame, the second fixations for free viewing (FV), the third gaze density maps for FV, the fourth fixations for the surveillance-viewing task (called Task), and the fifth gaze density maps for Task.

## 3.5. EyeTrackUAV2 in Brief

We have created a dataset containing binocular gaze information collected during two viewing conditions (free viewing and task) over 43 UAV videos (30 fps, 1280×720 and 720×480—42,241 frames, 1,408 s) observed by 30 participants per condition, leading to 1,239,157 fixations in free-viewing and 1,269,433 in task-viewing for dominant eyes positions. Notably, selected UAV videos show diversity in rendered environments, movement and size of objects, aircraft flight heights and angles to the ground, duration, size, and quality. This dataset overcomes the limitations of EyeTrackUAV1 in that it enables investigations of salience in more test sequences, on larger population samples, and for both free-viewing and task-based attention. Additionally, and even though they are still too few, three IR videos are part of visual stimuli.

Fixations, saccades, and gaze density maps were computed—for both eyes in additive and averaged fashions (see Binocular and BothEyes scenarios described later) and for the dominant eye—and are publicly available with original content and raw data on our FTP ftp://dissocie@ftp.ivc.polytech.univ-nantes.fr/EyeTrackUAV2/. The code in MATLAB to generate all ground truth information is also made available.

## 4. Analyses

In this section, we characterize the proposed EyeTrackUAV2 database. On one hand, we compare salience between six ground truth generation scenarios. This study can be beneficial to the community to know what is the potential error made when selecting a specific ground truth scenario over another. On the other hand, UAV videos induce new visual experiences. Consequently, observers exhibit different behaviors towards this type of stimuli. Therefore, we investigate whether the center bias, one of the main viewing tendencies [28], still applies to EyeTrackUAV2 content.

*4.1. Six Different Ground Truths*

The first question we address concerns the method used to determine the ground truth. In a number of papers, researchers use the ocular dominance theory in order to generate the ground truth. This theory relies on the fact that the human visual system favors the input of one eye over the other should binocular images be too disparate on the retinas. However, the cyclopean theory gains more and more momentum [81,82]. It alleges that vision processes approximate a central point between two eyes, from which an object is perceived. Furthermore, lately, manufacturers achieved major improvements in eye-tracking systems. They are now able to record and calibrate the positions of both eyes separately. This allows for exploring what best practices to create salient ground truth are [81–83].

When examining the mean of absolute error between eye positions of all scenarios, we have found a maximum value of about 0.6 degrees of visual angle. That value is rather small compared to the Gaussian kernel convolved on eye positions. Thus, we question whether selecting a ground truth scenario over another makes a significant difference for saliency studies. Consequently, we compared gaze density maps generated for the six scenarios defined below.

We proposed to evaluate the potential errors made when different methods for creating the ground truth are used. Note that the true position of the user gaze is not available. Accordingly, we need to run a cross-comparison between several well-selected and representative ground truths. Scenarios highly similar to all others are the ones that will make fewer errors. In such a context, the more scenarios are included, the more complete and reliable the study is.

We tested six methods, namely left (L), right (R), binocular (B), dominant (D), non-dominant (nD), and both eyes (BE). B corresponds to the average position between the left and right eyes and can be called version signal (see Equation (2)). BE includes the positions of both L and R eyes, and hence comprises twice as much information as other scenarios (see Equation (3)). nD has been added to estimate the gain made when using dominant eye information. The two visual attention conditions FV and Task were examined in this study. Illustrations of scenarios gaze density maps and fixations as well as methods comparisons are presented in Figure 5. Below is presented the quantitative evaluation.

$$\begin{cases} x_B & = \lfloor \frac{x_L + x_R}{2} \rfloor \\ y_B & = \lfloor \frac{y_L + y_R}{2} \rfloor \end{cases} \tag{2}$$

$$\begin{cases} x_{BE} & = x_L \cup x_R \\ y_{BE} & = y_L \cup y_R \end{cases} \tag{3}$$

We ran a cross-comparison on six well-used saliency metrics: correlation coefficient (CC) ($\in [-1, 1]$), similarity (SIM) ($\in [0, 1]$) the intersection between histograms of saliency, area under the curve (AUC) Judd and Borji ($\in [0, 1]$), normalized scanpath saliency (NSS) ($\in ]-\infty, +\infty[$), and information gain (IG) ($\in [0, +\infty[$), which measures on average the gain in information contained in the saliency map compared to a prior baseline ($\in [0, +\infty[$). We did not report Kullback–Leibler divergence (KL) ($\in [0, +\infty[$) as we favored symmetric metrics. Moreover, even though symmetric in absolute value, IG provides different scores depending on fixations under consideration. We thus compared scenarios for fixations of both methods, which leads to two IG measures. More details on metrics and metrics behaviors are given in [36,71,84]. Table 5 presents the results of measures when comparing gaze density maps of two scenarios.

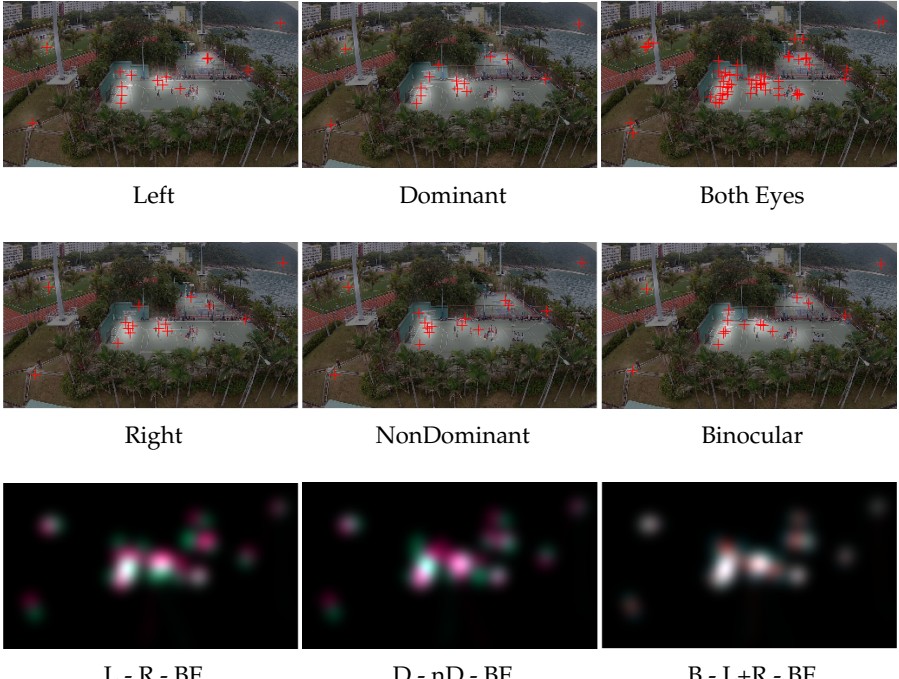

**Figure 5.** Qualitative comparison of gaze density maps for all scenarios on basketball, frame 401. Gaze density and fixations are displayed in transparency over the content. The last row compares scenarios: first scenario is attributed to the red channel, the second to green and the last to blue. When fully overlapping, the pixel turns white.

**Table 5.** Correlation coefficient (CC), similarity (SIM), and information gain (IG) results for scenarios cross-comparison. Red indicates the best scores and blue the least.

| SM1 | SM2 | CC ↑ | SIM ↑ | IG ↓ SM1-Fix1-SM2 | SM2-Fix2-SM1 | CC ↑ | SIM ↑ | IG ↓ SM1-Fix1-SM2 | SM2-Fix2-SM1 |
|---|---|---|---|---|---|---|---|---|---|
| | | | | **FV** | | | | **Task** | |
| Binocular | Dominant | 0.94 | 0.83 | 0.377 | 0.300 | 0.952 | 0.850 | 0.276 | 0.148 |
| Binocular | EyeNonDom | 0.95 | 0.84 | 0.370 | 0.301 | 0.952 | 0.849 | 0.283 | 0.163 |
| Binocular | Left | 0.94 | 0.83 | 0.371 | 0.301 | 0.948 | 0.843 | 0.264 | 0.192 |
| Binocular | Right | 0.94 | 0.83 | 0.390 | 0.324 | 0.944 | 0.838 | 0.304 | 0.152 |
| Binocular | BothEyes | **0.98** | **0.90** | 0.246 | **0.139** | **0.983** | **0.916** | 0.177 | **0.012** |
| Dominant | BothEyes | 0.96 | 0.87 | **0.158** | 0.374 | 0.967 | 0.873 | **0.143** | 0.248 |
| EyeNonDom | BothEyes | 0.97 | 0.87 | 0.167 | 0.394 | 0.966 | 0.872 | 0.144 | 0.228 |
| Left | BothEyes | 0.96 | 0.86 | 0.166 | 0.387 | 0.960 | 0.861 | 0.174 | 0.232 |
| Right | BothEyes | 0.96 | 0.86 | 0.181 | 0.416 | 0.960 | 0.862 | 0.147 | 0.279 |
| Dominant | EyeNonDom | 0.87 | 0.74 | 1.115 | 1.069 | 0.873 | 0.747 | 0.743 | 0.781 |
| Dominant | Left | 0.95 | 0.88 | 0.341 | 0.339 | 0.903 | 0.792 | 0.520 | 0.582 |
| Dominant | Right | 0.91 | 0.79 | 0.810 | 0.757 | 0.957 | 0.884 | 0.256 | 0.233 |
| EyeNonDom | Right | 0.96 | 0.88 | 0.346 | 0.342 | 0.902 | 0.793 | 0.587 | 0.519 |
| Left | EyeNonDom | 0.91 | 0.79 | 0.792 | 0.754 | 0.957 | 0.884 | 0.256 | 0.231 |
| Left | Right | **0.85** | **0.72** | **1.176** | **1.121** | **0.850** | **0.725** | **0.877** | **0.782** |
| Mean | | 0.937 | 0.832 | 0.467 | 0.488 | 0.938 | 0.839 | 0.343 | 0.319 |
| Std | | 0.037 | 0.052 | 0.340 | 0.295 | 0.038 | 0.053 | 0.230 | 0.234 |

Here are some insights extracted from the results:

- There was a high similarity between scenario gaze density maps. As expected, scores were pretty high (respectively low for IG), which indicates the high similarity between scenarios.
- All metrics showed the best results for comparisons including Binocular and BothEyes scenarios, the highest being the Binocular-BothEyes comparison.
- Left–Right and Dominant–NonDominant comparisons achieved worst results.
- It was possible to know the population main dominant eye through scenario comparisons (not including two eyes information). When describing the population, we saw that a majority of

left-dominant-eye subjects participated in the FV test, while the reverse happened for the Task experiment. This fact is noticeable in metric scores.

To verify whether scenarios present statistically significant differences, we have conducted an analysis of variance (ANOVA) on the scores obtained by the metrics. ANOVA results are presented in Table 6. All metrics showed statistically different results ($p < 0.05$) except for AUC Borji, AUC Judd, and NSS. It shows that, with regard to these three metrics, using a scenario over another makes no significant difference. This also explains why we did not report AUC and NNS results in Table 5. We further explore the other metrics, namely CC, SIM, and IG, through the Tukey's multiple comparison post-hoc test [85] ($\alpha = 0.05$).

**Table 6.** ANOVA analysis.

| | | FV | | Task | |
| | | F-Value | *p*-Value | F-Value | *p*-Value |
|---|---|---|---|---|---|
| $p < 0.05$ | CC | F(14,630) = 77.72 | $5 \times 10^{-127}$ | F(14,630) = 172.55 | $9 \times 10^{-205}$ |
| | SIM | F(14,630) = 200.07 | $5 \times 10^{-221}$ | F(14,630) = 309.43 | $2 \times 10^{-271}$ |
| | IG | F(14,630) = 158.96 | $5 \times 10^{-196}$ | F(14,630) = 156.16 | $4 \times 10^{-194}$ |
| $p > 0.05$ | AUCJ | F(14,630) = 0.36 | 0.9857 | F(14,630) = 0.4 | 0.9742 |
| | AUCB | F(14,630) = 0.22 | 0.9989 | F(14,630) = 0.05 | 1 |
| | NSS | F(14,630) = 0.95 | 0.5036 | F(14,630) = 0.92 | 0.5344 |

Results are presented in Figure 6. On the charts, we can see where stands group means and comparison intervals for each scenario over the entire dataset. Scenarios having non-overlapping intervals are statistically different.

In details, our results show three statistically distinct categories for all metrics.

- The first category showed the best similarities (i.e., highest for CC and SIM, and lowest for IGs) between scenarios, include all comparisons involving B and BE scenarios. There are also comparisons between single eye signals being the most and the least representative of the population's eyedness (e.g., D vs L and nD vs R for FV, D vs R and nD vs L for Task).
- The second category included the comparison of single eye signals that do not represent the "same" population's eyedness (i.e., D vs R and nD vs L for FV, D vs L and nD vs R for Task)
- Lastly, the third category presented the least similar scenario comparisons, in terms of the four evaluated metrics. Those comparisons are the single eye signals that come from different eyes (i.e., L vs R and D vs nD). Let us note that metrics gave reasonably good similarity scores, even for these two scenarios.

Also, there are some behaviors not common to all metrics that are worth mentioning. On one hand, for SIM and IG SM2–Fix2–SM1 in FV and Task, as well as for CC Task, scenario B vs. BE statistically achieved the best similarity scores. This strengthens even more the interest to use B or BE scenarios. On the other hand, for all metrics under the Task condition, and for SIM FV, the scenario L vs R was statistically different from D vs. nD. It also presented the least CC and SIM and the highest IG scores. This shows that under certain conditions, especially for Task attention in our context, D and nD scenarios may be favored over L and R signals.

Overall, over six metrics, three did not find significant differences between the scenarios' gaze density maps. The four others did, and indicated that using both eye information can be encouraged. Then, if not possible, eye-dominance-based signals may be favored over left and right eye scenarios, in particular under task-based attention. We stress out that overall, the difference between scenarios is rather small, as three metrics could not differentiate them.

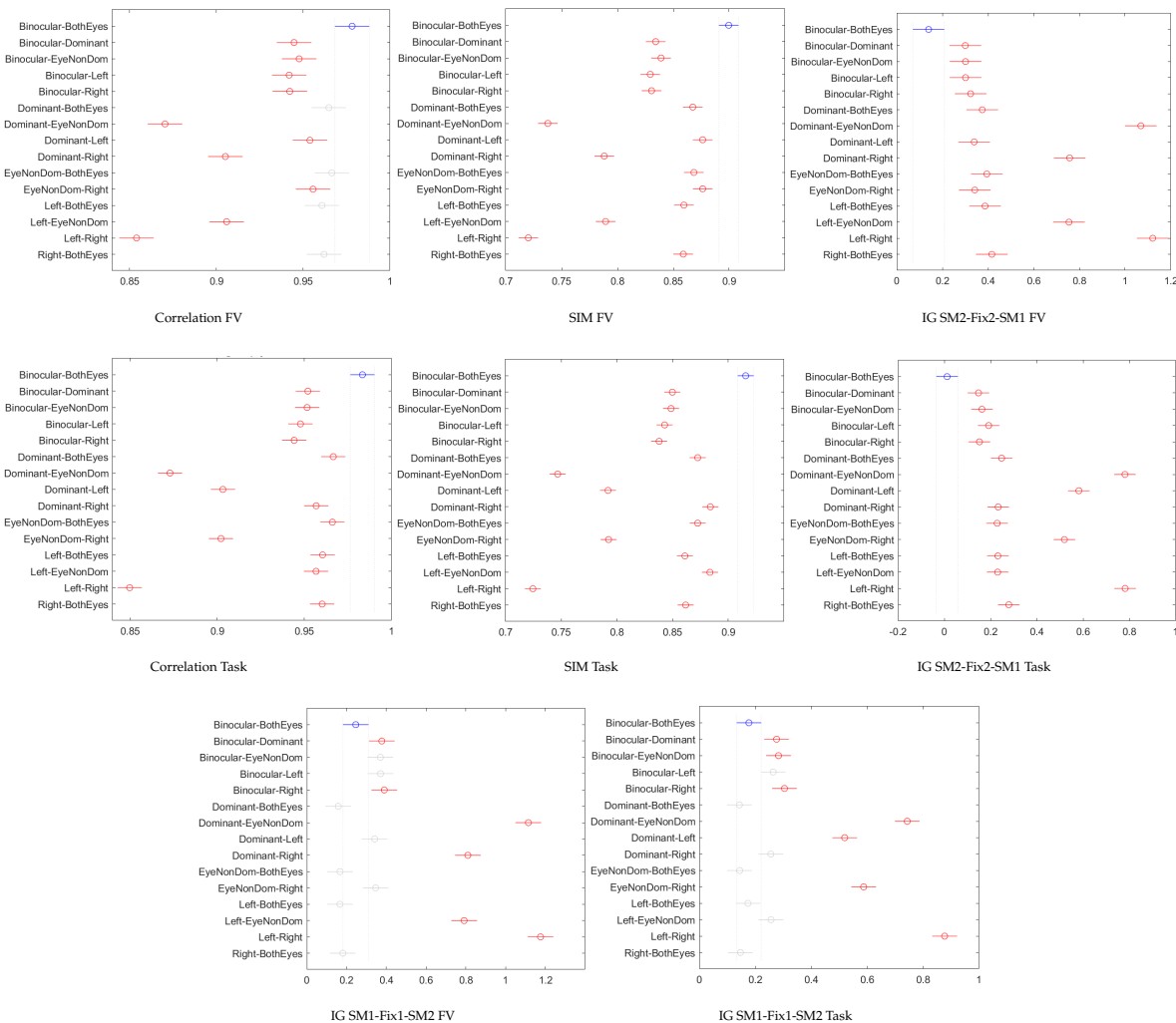

**Figure 6.** Tuckey's multiple comparison post-hoc test on scenarios for CC, SIM and IGs measures.

## 4.2. Biases in UAV Videos

The importance of the center bias in visual saliency for conventional imaging has been shown in Section 2. We wondered whether the center bias is systematically present in UAV content. This section aims to shed light on this question, qualitatively and quantitatively.

### 4.2.1. Qualitative Evaluation of Biases in UAV Videos

We evaluated the viewing tendencies of observers thanks to the average gaze density map, computed over the entire sequence. It was representative of the average position of gaze throughout the video. It was used to observe potential overall biases, as it could be the case with the center bias. Figures 7 and 8 show the average gaze density map for all sequences of *EyeTrackUAV2* dataset, generated from D scenario, for both free-viewing and task-viewing conditions. Several observations can be made.

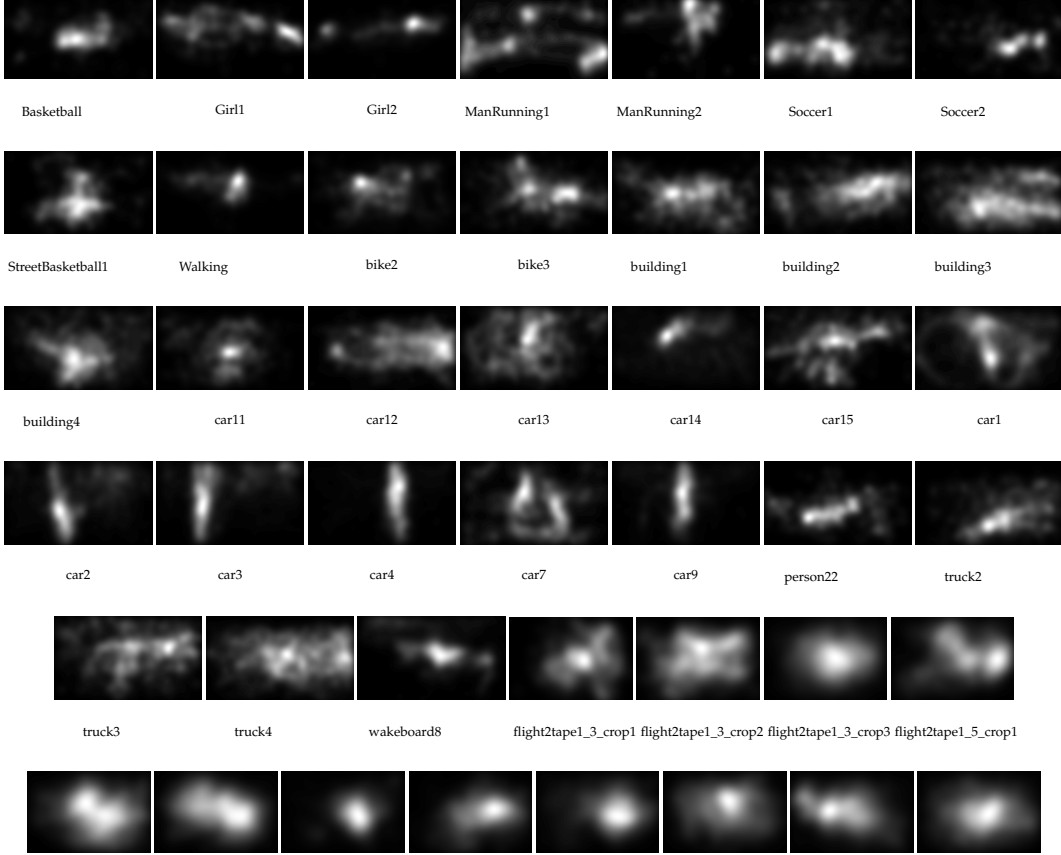

**Figure 7.** Average gaze density maps for all sequences of *EyeTrackUAV2* dataset, generated from D scenario, for the free-viewing condition.

Content-dependent center bias. We verify here the content-dependence of the center bias in UAV videos. For both attention conditions, the scene environment and movements exacerbates or not UAV biases. For instance, in sequences car 2–9 (fourth row), the aircraft is following cars on a road. Associated average gaze density maps display the shape of the road and its direction, i.e., vertical route for all and roundabout for car7. Car 14 (third row), a semantically similar content except that it displays only one object on the road with a constant reframing (camera movement) which keeps the car at the same location, presents an average gaze density map centered on the tracked object.

Original database-specific center bias. We can observe that a center bias was present in VIRAT sequences, while videos from other datasets, namely UAV123 and DTB70, did not present this bias systematically. The original resolution of content and the experimental setup are possibly the sources of this result. Indeed, the proportion of content seen at once was not the same for all sequences: 1.19% of a VIRAT content was seen per degree of visual angle, whereas it was 0.44% for the two other original databases. VIRAT gaze density maps were thus smoother, which resulted in higher chances to present a center bias. To verify this assumption based on qualitative assessment, we computed the overall gaze density maps for sequences coming from each original dataset, namely DTB70, UAV123, and VIRAT. These maps are shown in Figure 9. VIRAT gaze density maps are much more concentrated and centered. This corroborates that biases can be original-database-specific.

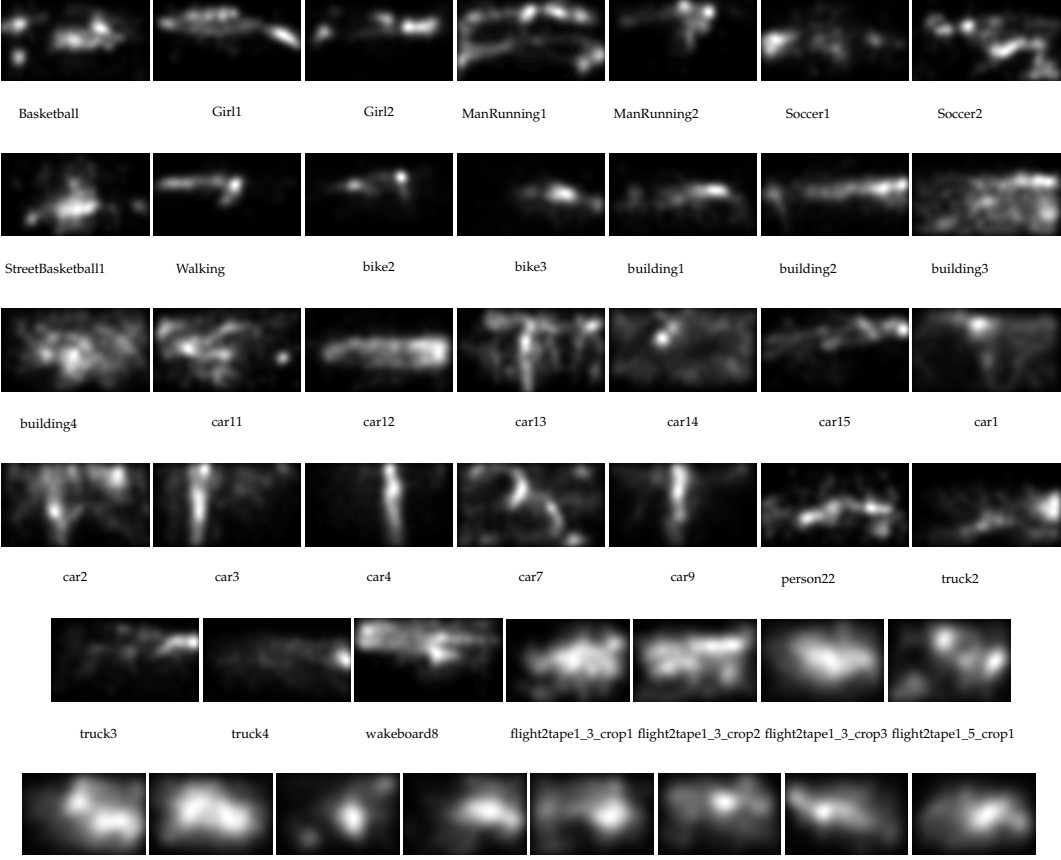

**Figure 8.** Average gaze density maps for all sequences of *EyeTrackUAV2* dataset, generated from D scenario, for the task-viewing condition.

Task-related gaze density maps seemed more spread out. Task-based gaze density maps covered more content when compared to free-viewing condition for most sequences (e.g., in about 58% of videos such as basketball, car11, car2, and wakeboard). This behavior is also illustrated in Figure 9. We correlate this response with the object detection task. Visual search scanning implies an extensive exploration of the content. However, 21% of the remaining sequences (i.e., soccer1, bike2–3, building 1–2, car1 and 15, and truck3–4) show less discrepancies in the task-viewing condition than in free-viewing condition. We do not find correlation between such behavior and sequences characteristics given in Table 3. This leaves room for further exploration of differences between task-based and free viewing attention.

Overall, there was no generalization of center bias for UAV content. As stated earlier, we do not observe a systematic center bias, except for VIRAT sequences. This is especially true for task-related viewing. However, we observe specific patterns. Indeed, vertical and horizontal potato-shaped salient areas are quite present in average gaze density maps of EyeTrackUAV2. Such patterns are also visible in UAV2 and DTB70 overall gaze density maps, especially in task-viewing condition. This indicates future axes of developments for UAV saliency-based applications. For instance, instead of using a center bias, one may introduce priors as a set of prevalent saliency area shapes with different directions and sizes [86].

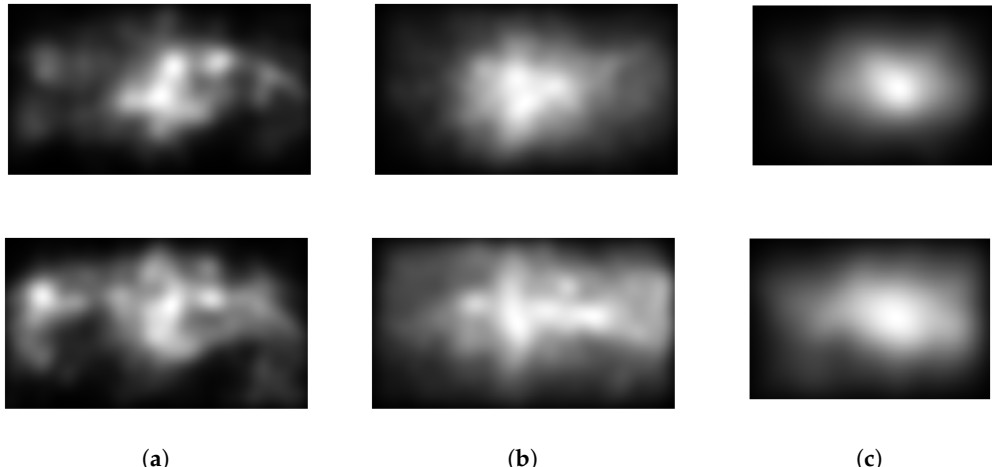

(**a**)                                  (**b**)                                  (**c**)

**Figure 9.** Overall average gaze density maps per original dataset, generated from dominant (D) scenario, in free-viewing (top-row) and Task-viewing (bottom row) for original datasets: (**a**) DTB70; (**b**) UAV123; (**c**) VIRAT.

### 4.2.2. Quantitative Evaluation of the Central Bias in UAV Videos

To go further into content-dependencies, we investigated quantitatively the similarity of dominant-eye-generated gaze density maps with a pre-defined center bias. Figure 10 presents the center bias baseline created in this purpose as suggested in [35,36] .



**Figure 10.** Center prior baseline.

We performed the evaluation based on four well-used saliency metrics: CC, SIM, KL, and IG. Results are presented in Table 7. They support the observations we made in the previous section. Overall scores do not reveal a high similarity with the center prior (e.g., maximum CC and SIM of about 0.5, high KL and IG). On the other hand, we observe content-specific center prior in UAV123 and DTB70. For instance, videos more prone to center bias includes sequences extracted from VIRAT and *building1,3,4*, and *car13*. On the contrary, sequences *Girl1-2, ManRunning1-2, Walking, car4*, and *wakeboard8* are not likely to present center bias. This confirms there is no generalization of center bias for UAV content. Regarding differences between free-viewing and task-viewing conditions, results are inconclusive as no systematic behavior is clearly visible from this analysis.

### 5. Conclusions

UAV imaging modifies the perceptual clues of typical scenes due to its bird point of view, the presence of camera movements and the high distance and angle to the scene. For instance, low-level visual features, and size of objects change and depth information is flattened or disappears (e.g., presence of sky). To understand observers' behaviors toward these new features, especially in terms of visual attention and deployment, there is a need for large-scale eye-tracking databases for saliency in UAV videos. This dataset is also a key factor in the field of computational models of visual attention, in which large scale datasets are required to train the latest generation of deep-based models.

**Table 7.** Comparison of gaze density maps with the center bias presented in Figure 10. Are displayed in red the numbers over (or under for Kullback–Leibler divergence (KL) and IG) measures average, indicated in the last row.

| | FV | | | | Task | | | |
|---|---|---|---|---|---|---|---|---|
| | CC ↑ | SIM ↑ | KL ↓ | IG ↓ | CC ↑ | SIM ↑ | KL ↓ | IG ↓ |
| VIRAT_09152008flight2tape1_3_crop1 | **0.50** | **0.48** | **7.17** | **1.53** | **0.46** | **0.48** | **6.85** | **1.62** |
| VIRAT_09152008flight2tape1_3_crop2 | **0.49** | **0.52** | **5.59** | **1.50** | **0.36** | **0.48** | **6.42** | **1.75** |
| VIRAT_09152008flight2tape1_3_crop3 | **0.46** | **0.43** | **8.46** | **1.91** | **0.37** | **0.43** | **7.98** | **1.99** |
| VIRAT_09152008flight2tape1_5_crop1 | 0.27 | **0.38** | **9.77** | **2.29** | 0.18 | **0.36** | **10.14** | **2.49** |
| VIRAT_09152008flight2tape1_5_crop2 | **0.42** | **0.44** | **8.05** | **1.90** | **0.30** | **0.45** | **7.41** | **1.87** |
| VIRAT_09152008flight2tape2_1_crop1 | **0.41** | **0.39** | **9.34** | **2.05** | **0.38** | **0.42** | **8.55** | **1.97** |
| VIRAT_09152008flight2tape2_1_crop2 | **0.40** | **0.35** | **10.90** | **2.50** | **0.32** | **0.42** | **8.01** | **2.01** |
| VIRAT_09152008flight2tape2_1_crop3 | **0.42** | **0.40** | **9.46** | **2.11** | **0.28** | **0.39** | **9.30** | **2.24** |
| VIRAT_09152008flight2tape2_1_crop4 | **0.36** | **0.36** | **10.35** | **2.34** | **0.28** | **0.38** | **9.79** | **2.30** |
| VIRAT_09152008flight2tape3_3_crop1 | **0.42** | **0.43** | **8.16** | **1.96** | **0.35** | **0.43** | **7.84** | **2.03** |
| VIRAT_09162008flight1tape1_1_crop1 | **0.47** | **0.45** | **7.76** | **1.80** | **0.37** | **0.42** | **8.40** | **2.00** |
| VIRAT_09162008flight1tape1_1_crop2 | **0.40** | **0.40** | **9.14** | **2.14** | **0.27** | **0.40** | **8.91** | **2.22** |
| UAV123_bike2 | **0.39** | **0.34** | **11.51** | **2.43** | **0.34** | 0.29 | 13.21 | 2.82 |
| UAV123_bike3 | **0.39** | **0.34** | **11.71** | **2.37** | **0.29** | 0.26 | 14.34 | 2.96 |
| UAV123_building1 | **0.40** | **0.37** | **10.64** | **2.18** | **0.32** | 0.31 | 12.74 | **2.69** |
| UAV123_building2 | 0.30 | **0.33** | **11.89** | **2.43** | 0.18 | 0.27 | 13.87 | 3.06 |
| UAV123_building3 | 0.27 | **0.34** | **11.50** | **2.42** | 0.17 | **0.32** | **11.82** | **2.56** |
| UAV123_building4 | **0.39** | **0.36** | **10.82** | **2.20** | **0.35** | **0.39** | **9.72** | **2.10** |
| UAV123_car11 | **0.37** | **0.32** | 12.37 | **2.58** | 0.21 | 0.30 | 12.68 | **2.67** |
| UAV123_car12 | 0.21 | 0.28 | 13.35 | 2.80 | **0.26** | 0.29 | 13.12 | 2.69 |
| UAV123_car13 | 0.30 | **0.34** | **11.48** | **2.39** | 0.20 | **0.33** | **11.50** | **2.44** |
| UAV123_car14 | 0.20 | 0.25 | 14.47 | 3.16 | 0.12 | 0.31 | 12.28 | 2.71 |
| UAV123_car15 | **0.31** | **0.34** | **11.52** | **2.47** | 0.10 | 0.30 | 12.70 | 2.81 |
| UAV123_car1 | 0.21 | 0.26 | 14.33 | 3.10 | 0.13 | 0.30 | 12.61 | 2.77 |
| UAV123_car2 | 0.22 | 0.27 | 13.91 | 3.02 | 0.13 | 0.30 | 12.68 | 2.80 |
| UAV123_car3 | 0.16 | 0.24 | 14.77 | 3.19 | 0.14 | 0.28 | 13.39 | 2.93 |
| UAV123_car4 | 0.22 | 0.20 | 16.27 | 3.55 | 0.20 | 0.24 | 14.76 | 3.23 |
| UAV123_car7 | 0.22 | 0.23 | 15.11 | 3.16 | 0.11 | 0.28 | 13.13 | 2.92 |
| UAV123_car9 | 0.26 | 0.23 | 15.41 | 3.27 | 0.21 | 0.28 | 13.69 | 2.86 |
| UAV123_person22 | **0.35** | 0.31 | 12.44 | **2.60** | **0.27** | 0.31 | 12.45 | 2.68 |
| UAV123_truck2 | 0.27 | **0.32** | 12.29 | **2.56** | 0.09 | 0.27 | 13.66 | 3.01 |
| UAV123_truck3 | 0.27 | **0.35** | **11.14** | **2.34** | 0.12 | 0.31 | 12.23 | 2.73 |
| UAV123_truck4 | 0.29 | **0.36** | **10.71** | **2.34** | 0.16 | 0.29 | 13.18 | 3.03 |
| UAV123_wakeboard8 | 0.23 | 0.21 | 15.91 | 3.45 | 0.11 | 0.24 | 14.93 | 3.29 |
| DTB70_Basketball | **0.38** | 0.27 | 14.13 | 2.89 | **0.30** | 0.31 | 12.30 | **2.59** |
| DTB70_Girl1 | 0.16 | 0.28 | 13.47 | 2.90 | 0.15 | 0.25 | 14.54 | 3.18 |
| DTB70_Girl2 | 0.20 | 0.20 | 16.04 | 3.60 | 0.19 | 0.23 | 15.04 | 3.34 |
| DTB70_ManRunning1 | 0.02 | 0.16 | 17.45 | 4.09 | 0.00 | 0.20 | 16.11 | 3.73 |
| DTB70_ManRunning2 | 0.12 | 0.13 | 18.40 | 4.31 | 0.10 | 0.15 | 17.99 | 4.24 |
| DTB70_Soccer1 | 0.21 | 0.26 | 14.23 | 3.04 | 0.17 | 0.26 | 14.03 | 3.18 |
| DTB70_Soccer2 | 0.21 | 0.22 | 15.56 | 3.33 | 0.22 | **0.32** | **11.86** | **2.69** |
| DTB70_StreetBasketball1 | **0.33** | 0.26 | 14.29 | 2.94 | **0.28** | 0.26 | 14.29 | 3.00 |
| DTB70_Walking | 0.29 | 0.20 | 16.14 | 3.51 | **0.27** | 0.22 | 15.81 | 3.51 |
| mean | 0.31 | 0.32 | 12.27 | 2.67 | 0.23 | 0.32 | 12.01 | 2.69 |

This need is even stronger with the fast expansion of applications related to UAV, for leisure and professional civilian activities and a wide range of military services. Combining UAV imagery with one of the most dynamic research fields in vision, namely salience, is highly promising, especially for videos that are gaining more and more attention these last years.

This work addresses the need for such a dedicated dataset. An experimental process has been designed in order to build a new dataset, EyeTrackUAV2. Gaze data were collected during the observation of UAV videos under controlled laboratory conditions for both free viewing and

object-detection surveillance task conditions. Gaze positions have been collected on 30 participants for each attention condition, on 43 UAV videos in 30 fps, 1280×720 or 720×480, consisting of 42,241 frames and 1408 s. Overall, 1,239,157 fixations in free-viewing and 1,269,433 in task-viewing were extracted from the dominant eye positions. Test stimuli were carefully selected from three original datasets, i.e., UAV123, VIRAT, and DTB70, to be representative as much as possible of the UAV ecosystem. Accordingly, they present variations in terms of environments, camera movement, size of objects, aircraft flight heights and angles to the ground, video duration, resolution, and quality. Also, three sequences were recorded in infra-red.

The collected gaze data were analyzed and transformed into fixation and saccade eye movements using an I-DT-based identification algorithm. Moreover, the eye-tracking system high frequency of acquisition enabled the production of gaze density maps for each experimental frame of the examined video stimuli directly from raw data. The dataset is publicly available and includes, for instance, raw binocular eye positions, fixation, and gaze density maps generated from the dominant eye and both eyes information.

We further characterized the dataset considering two different aspects. On one hand, six scenarios, namely binocular, both eyes, dominant eye, non-dominant eye, left, and right can be envisioned to generate gaze density maps. We wondered whether a scenario should be favored over another or not. Comparisons of scenarios have been conducted on six typical saliency metrics for gaze density maps. Overall, all scenarios are pretty similar: over the six evaluated metrics, three could not make a distinction between scenarios. The three last metrics present mild but statistically significant differences. Accordingly, the information of both eyes may be favored to study saliency. If not possible, choosing information from the dominant eye is encouraged. This advice is not a strict recommendation. On the other hand, we notice that conventional biases in saliency do not necessarily apply to UAV content. Indeed, the center bias is not systematic in UAV sequences. This bias is content-dependent as well as and task-condition-dependent. We observed new prior patterns that must be examined in the future.

In conclusion, the EyeTrackUAV2 dataset enables in-depth studies of visual attention through the exploration of new salience biases and prior patterns. It establishes in addition a solid basis on which dynamic salience for UAV imaging can build upon, in particular for the development of deep-learning saliency models.

**Author Contributions:** Data curation, V.K.; formal analysis, A.-F.P.; supervision, L.Z., V.R., M.P.D.S., and O.L.M.; Writing—original draft, A.-F.P.; Writing—review and editing, A.-F.P. and O.L.M. All authors have read and agreed to the published version of the manuscript.

**Funding:** Agence Nationale de la Recherche: ANR-17-ASTR-0009.

**Acknowledgments:** The presented work is funded by the ongoing research project ANR ASTRID DISSOCIE (Automated Detection of SaliencieS from Operators' Point of View and Intelligent Compression of DronE videos) referenced as ANR-17-ASTR-0009. Specifically, the LS2N team ran the experiment, created and made available the *EyeTrackUAV2* dataset. The Univ Rennes team added binocular and both-eyed scenarios information to the dataset, conducted analyses, and reported it.

**Conflicts of Interest:** The authors declare no conflict of interest.

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
