# Peer review of "EyeTrackUAV2: A Large-Scale Binocular Eye-Tracking Dataset for UAV Videos"

_drones, doi:10.3390/drones4010002_

Round 1

Reviewer 1 Report

Thank you for your nice work. The study is interesting, however, it is not fit for journal publication at its current form. Therefore, I would kindly suggest the following remarks are addressed before moving on with comments/ suggestion.

1.       The abstract should include how the authors collect and process the datasets, i.e., the methodology. The background descriptions (sentence 1-4) in the abstract can be reduced/ shortened.

2.       Some studies have been carried out on visual salience and biological motion interaction. It would be more interesting if the authors can include a paragraph in the introduction to discuss why this work is necessary by referring to some references in:

Tad T. Brunye, Shaina B. Martis, Carlene Horner, John A. Kirejczyk, Kathryn Rock, Visual salience and biological motion interact to determine camouflaged target detectability, Applied Ergonomics, Volume 73, 2018, Pages 1-6, ISSN 0003-6870

3.       The locations of the table captions are inconsistent. For example, the caption of Table 2 is on top of the table, while the caption of Table 3 is on the bottom.

4.       Is there any connection between Table 4 and Figure 3 which are presented next to each other?

5.       The equations shown in Subsection 4.1.1. are not indexed.

6.       The authors mentioned in the article that the datasets (original content, raw data, saliency data) are publicly available. However, the EyeTrackUAV2 folder in the given link is empty.

7.       This manuscript is not easy to read. For example, some paragraphs are too short (contain 1 or 2 sentences only), and some sentences are too long (line 18, etc.). The authors should restructure the manuscript.

8.       Furthermore, some sections in the manuscript (e.g., Section 4) are too lengthy, and their content makes the paper read like a technical report.

9.       It is also noticeable that the descriptions of other literature are quite extensive, especially in subsection 4.2 (it may happen in other sections). It would be better to reduce them or move them to the related work section.

10.   Since the results presented in this paper are only the datasets, it would be better if the authors can share the possible works using these datasets and their applications in the conclusion.

Author Response

First, we would like to thank the reviewer for his/her time and interesting comments on our contribution. We acknowledge that thanks to that, the readability and overall quality of the paper have increased. 

We give our answers below (in black), following each comment made by the reviewer.

Comments and Suggestions for Authors

Thank you for your nice work. The study is interesting, however, it is not fit for journal publication at its current form. Therefore, I would kindly suggest the following remarks are addressed before moving on with comments/ suggestion.

We are glad you appreciate our work. We did our best to address the issues you pointed out to fit the journal publication.

The abstract should include how the authors collect and process the datasets, i.e., the methodology. The background descriptions (sentence 1-4) in the abstract can be reduced/ shortened.

This is a good point. Thank you for the notice. We have slightly reduced the background description and added most important details related to the dataset in the new version: “It consists of the collection of precise binocular gaze information (1000 Hz) over 43 videos (RGB, 30 fps, 1280x720 or 720x480). Thirty participants observed stimuli under both free viewing and task conditions. Fixations and saccades were then computed with the I-DT algorithm, while gaze density maps were calculated by filtering eye positions with a Gaussian kernel.

Some studies have been carried out on visual salience and biological motion interaction. It would be more interesting if the authors can include a paragraph in the introduction to discuss why this work is necessary by referring to some references in:

Tad T. Brunye, Shaina B. Martis, Carlene Horner, John A. Kirejczyk, Kathryn Rock, Visual salience and biological motion interact to determine camouflaged target detectability, Applied Ergonomics, Volume 73, 2018, Pages 1-6, ISSN 0003-6870

You are referring to a very interesting study. Thank you for bringing our attention to this paper and its field. We have included the reference in related work (Section 2, 3§), as it perfectly illustrates and addresses the interest on anomaly detection in military applications while emphasizing the need for dynamic salience studies.

The locations of the table captions are inconsistent. For example, the caption of Table 2 is on top of the table, while the caption of Table 3 is on the bottom.

This is very accurate. We have corrected this inaccuracy in the new version.

Is there any connection between Table 4 and Figure 3 which are presented next to each other?

There is no relation or connection between Table 4 and Figure 3. This is why they have separate captions and no overall one. As we think their resolution is correct in the present layout, we have not changed the configuration of these table and figure.

The equations shown in Subsection 4.1.1. are not indexed.

This is correct indeed. Thank you for notifying us. We have corrected this issue in the new version of the paper.

The authors mentioned in the article that the datasets (original content, raw data, saliency data) are publicly available. However, the EyeTrackUAV2 folder in the given link is empty.

The link in the first version was only informative and was not suited for accessing the FTP. We should have notified this in the cover letter to let you know. Sorry for the inconvenience. In addition, the database is not open to the public yet. We think it is reasonable to make it public only when the paper is accepted.

We here provide you the credentials for accessing the private FTP so that you can fully review our work. Here is the information you need:

The correct address to connect to the private FTP is: ftp://[email protected]/EyeTrackUAV2/

The user id is:            dissocie

And the password:     KOo1NVlg

This manuscript is not easy to read. For example, some paragraphs are too short (contain 1 or 2 sentences only), and some sentences are too long (line 18, etc.). The authors should restructure the manuscript.

This is unfortunate. We have put a lot of effort to make it as clear as possible. We have gone through massive revisions and restructured the paper to make it readable.

Furthermore, some sections in the manuscript (e.g., Section 4) are too lengthy, and their content makes the paper read like a technical report.

We realized it after your comment. Thank you for noting this. We actually have drastically changed this section. We have removed the part on MAE that was mildly beneficial to the reader. We have added more information regarding the statistical analysis. The new version should convey more information that is useful, clearer and to the point.

It is also noticeable that the descriptions of other literature are quite extensive, especially in subsection 4.2 (it may happen in other sections). It would be better to reduce them or move them to the related work section.

Thank you for your comment. Accordingly, we have included the section 2 “related work” that embraces not only previous datasets for salience but also all information related to visual attention, saliency, and attentional biases, among others. This makes the present version of the paper comprehensive regarding the state of the art, well balanced and organized.

Since the results presented in this paper are only the datasets, it would be better if the authors can share the possible works using these datasets and their applications in the conclusion.

This is an interesting perspective. We have a slightly different view on this point. We would like to focus at the end of the conclusion on the results obtained in the analysis, section 4. Indeed, the recommendation for ground truth generation is important for the community. Moreover, in our opinion, what is even more compelling is calling into question the presence of center bias in UAV video salience. We have consequently chosen not to add more content to the conclusion.

Please consider that we have described potential applications in the abstract and the introduction while giving more details on applications in the related work section.

We would like to thank again the reviewer for his/her relevant comments. We made our best to address the pointed issues.

Reviewer 2 Report

Summary:

The present study aimed at providing a dataset of eye-movements on Unmanned Aerial Vehicles (UAVs). This dataset is populated with videos originating from 3 different datasets. The videos provided different angles and points of view as well as the content differed amongst them. Authors recorded eye-movements performed on each videos of the dataset in two different tasks: free-viewing and surveillance. Then, they build fixation maps in order to evaluate differences between different ways to record/use eye movements data. Analysis of eye-movements provided guidelines for the generation of saliency ground truth. Indeed, the use of both eyes or binocular recording was found to provide the best results. Moreover, authors analyzed central bias and showed that the difference in task did not impact central bias. However, the central bias was more present in a specific dataset. Altogether, this study provide guideline to develop saliency models for UAVs similarly to what have been done in the MIT benchmark.

Broad comments:

Authors should discuss the differences between models of saliency used for either bottom-up or top-down. These two approaches are completely different and top-down saliency models is by far one of the most challenging issue in eye-movements prediction. At the end of page 2, authors highlight the need for dynamic saliency models. However, such models exists already (e.g. Targino Da Costa, A. L. N., & Do, M. N. (2014). A retina-based perceptually lossless limit and a gaussian foveation scheme with loss control. IEEE Journal on Selected Topics in Signal Processing, 8(3), 438–453, doi.org/10.1109/JSTSP.2014. 2315716. For more models have a look to the MIT saliency benchmark). Thus, I expect authors to discuss why such model does not fit UAVs video content or to evaluate the need of dynamic saliency models tailored for UAV content with an evaluation of existing dynamic saliency models on UAVs videos. Other indices in the content selection such as number of object present or environment’s characteristic such as land, city etc. could have been used to describe the dataset more. I was also wondering why the VIRAT dataset was not excluded as due to the kept pixel size of the videos, comparison between VIRAT and the two other dataset would require specific and somehow difficult computation. I would like to invite authors to ensure that the differences in number of point in calibration does not produce any substantial differences in error as this could reassure the reader that such change did not have any impact over the quality of data required. The terminology « saliency maps » was used to designate convolved fixation maps. For clarity sake, I invite authors to change the terminology and refers to their computations as fixations maps instead of saliency maps as saliency maps refers to the maps extracted after computing the saliency from a saliency model. Moreover, any usage of the word saliency that relates to gaze or fixations should be clarified. « Salience ground truth » can remain unchanged. In general, the Analyses section should be reworked in order to provide more details about the methodology used (see specific comments). I am questioning the usefulness of comparing left and right eye at the same time of Dominant and Non-dominant as the dominant eye tracking is the common method used in eye tracking studies. Moreover, evaluating both sides and dominance make Table 5 to be redundant. Thus, in fine differences in average measurements for sides and dominance is the same and thus for the present purpose does not seems mandatory. Moreover this would help the ease to read analysis part. When running ANOVA and multiple comparisons, authors should provide the entire statistical information for principal effects and interaction effects (F(X,X)=XX.XX,p<.05). Afterwards multiple comparison is usually presented by comparing mean of the two compared group and indicating whether this effect is significant (p<.05) or not. All reference to p value below another threshold than .05 should be avoided (p<.10, p<.01, p<.001). Indeed such indication mislead to believe that p value indicates an effect size. I highly encourage authors to present their statistic in a more detailed way in order for the scientific community to evaluate the data analysis and the correctness of the statistics. I was really astonished to see that authors « discard » the non-significant metrics such as NSS. Indeed, NSS is now used by the MIT benchmark to sort model performance and is a very robust metrics (see Z Bylinskii, T Judd, A Oliva, A Torralba, F Durand What do different evaluation metrics tell us about saliency models? arXiv preprint arXiv:1604.03605, 2016; M Kammerer, T Wallis, M Bethge Information-theoretic model comparison unifies saliency metrics PNAS, 112(52), 16054-16059, 2015). Hence, emphasis over significant results should be decrease in the light of other metrics that were found not to be significant and that are a ground truth in the field of saliency evaluation. Authors have to rework their results interpretations and the conclusion see specific comments. As their analysis do not support whether recording one eye is better than binocular recording. Indeed, comparison in the strength of correlation indicates only that the correlation between binocular and two eye is significantly higher than other correlation comparisons. Thus authors should make sure that their analysis allow to draw such conclusions. Results should be rewrite to something like: « the correlation between binocular and both eyes recording is higher than both eye separately ». If authors want to claim that both eyes should be recorded then they must compare their condition using an ANOVA on a metric of performance of ocular movement on saliency and not on the correlation statistic. One way to increase authors’ saliency ground truth would be to enrich it with a infinite human observer as described in Judd, Durand, and Torralba 2012. The evaluation of the differences in eye movements between their two conditions (free viewing and surveillance) would have been a plus. This could have been performed using either grid segmentation/region of interest of the image or this one (Lao, J., Miellet, S., Pernet, C., Sokhn, N., & Caldara, R. (2017). iMap4: An open source toolbox for the statistical fixation mapping of eye movement data with linear mixed modeling. Behavior research methods, 49(2), 559-575). It could have been interesting to evaluate EM according to the complexity of video and thus modulate ground truth with indices such as the ones used in the paper.

Specific comments:

L33 & P16 L469: Citation with XXX and colleagues [10] should be favored as "In [10]"

L45: Should be simplified by where point eyes or eyes fixates. Delete « more » in the second sentence.

P5 Table 1 is presented before the presentation of TI and SI indices. Hence, either these indices should be describe in table caption or the table should appear after indices explanation.

P6 Statistics (mean, std…) for SI and TI indices could have been provided in table 2

L305-308: This paragraph is not clear to me. Either clarify or delete it.

L361-362: How did you do that ? Please develop the methodology used here.

L437-444: The formulation while presenting the correlation analysis should be rewrite. Indeed,« The worst scenario, significantly in Task » mislead that authors’ correlation analysis could provide evaluation the best scenario whereas it only indicate whether two metrics were correlated.

Author Response

First, we would like to thank the reviewer for his/her time and interesting comments on our contribution. This contributes to increase the quality of the paper and the way we reported our work. 

We give our answers below (in black), following each comment made by the reviewer.

Comments and Suggestions for Authors

Summary:

The present study aimed at providing a dataset of eye-movements on Unmanned Aerial Vehicles (UAVs). This dataset is populated with videos originating from 3 different datasets. The videos provided different angles and points of view as well as the content differed amongst them. Authors recorded eye-movements performed on each videos of the dataset in two different tasks: free-viewing and surveillance. Then, they build fixation maps in order to evaluate differences between different ways to record/use eye movements data.

In reality, we did not compute fixation maps. However, the vocabulary used was probably misleading. You can find more details in answers below.

Analysis of eye-movements provided guidelines for the generation of saliency ground truth. Indeed, the use of both eyes or binocular recording was found to provide the best results. Moreover, authors analyzed central bias and showed that the difference in task did not impact central bias.

The difference between Task and FV were not the focus of the study. Moreover, in the bias analysis section, results were inconclusive regarding our context of evaluation. We realized here that some re-writing were necessary for clarity purposes.

However, the central bias was more present in a specific dataset. Altogether, this study provide guideline to develop saliency models for UAVs similarly to what have been done in the MIT benchmark.

Broad comments:

Authors should discuss the differences between models of saliency used for either bottom-up or top-down. These two approaches are completely different and top-down saliency models is by far one of the most challenging issue in eye-movements prediction.

Thank you for providing this very good comment. The difference and relation between bottom-up and top-down attention are indeed complex. In the paper, we refer to two works that nicely cover this issue:

Katsuki, F.; Constantinidis, C.; Bottom-up and top-down attention: different processes and overlapping neural systems; The Neuroscientist; 2014

Krasovskaya, S.; MacInnes, W.J.; Salience Models: A Computational Cognitive Neuroscience Review; Vision; 2019

We believe that commenting further on the issue will lead the reader slightly out of the scope of our study. For this reason, we did not add more information to the question, even though this issue is important and interesting for the community.

At the end of page 2, authors highlight the need for dynamic saliency models. However, such models exists already (e.g. Targino Da Costa, A. L. N., & Do, M. N. (2014). A retina-based perceptually lossless limit and a gaussian foveation scheme with loss control. IEEE Journal on Selected Topics in Signal Processing, 8(3), 438–453, doi.org/10.1109/JSTSP.2014. 2315716. For more models have a look to the MIT saliency benchmark). Thus, I expect authors to discuss why such model does not fit UAVs video content or to evaluate the need of dynamic saliency models tailored for UAV content with an evaluation of existing dynamic saliency models on UAVs videos.

Thank you for noting this. It makes sense to wonder if this dataset is useful. We have studied the efficiency of current handcrafted and deep models of salience, static and dynamic, in a previous study. We wanted to know, as you wondered here, if there is a need for new models for UAV videos. Is this imaging so specific that current solutions are not suited for it?

Our results showed that deep solutions were the most efficient models (based on usual metrics for benchmarking, i.e. these of the MIT saliency benchmark). However, this efficiency is not reaching the expectations we had nor the efficiencies of such models on conventional imaging. We attributed this behavior to the modification of low-level features in UAV videos. Besides, there is room for improvement regarding the temporal dimension. You can find the details in our paper:

Perrin, A.F.; Zhang, L.; Le Meur; O. How well current saliency prediction models perform on UAVs videos? International Conference on Computer Analysis of Images and Patterns; Springer; 2019

This information clearly deserves to be included in the paper. It indeed justifies the need for our dataset. Accordingly, we completed and stressed out the information about this work. Such details are included in related work, Section 2.

Other indices in the content selection such as number of object present or environment’s characteristic such as land, city etc. could have been used to describe the dataset more.

Thank you for rising this challenging point. Having more information about the content is always interesting to characterize a dataset. We have included five important features to describe UAV videos. Among which is the environment. We have selected the vocabulary "urban", "suburban", and "rural" to characterize it (See Table 2). “Land” and “city” belong to these annotations.

We agree that some more insights may be helpful. For instance the number of objects and camera movements. However, counting the number of objects in such content can be fastidious and complex. Moreover, differences in size and semantic may influence human behaviors towards objects. Only including the number of objects is thus probably useless. Including multiple representations that are more sophisticated generates costs that seem not worth it in our case.

Regarding camera movements, it is quite hard to decide how to characterize camera movements. Variations are time-dependent and due to the UAV present numerous degrees of freedom. For these reasons, we decided not to characterize further the videos.

I was also wondering why the VIRAT dataset was not excluded as due to the kept pixel size of the videos, comparison between VIRAT and the two other dataset would require specific and somehow difficult computation.

Thank you for this very interesting question. The main advantage of VIRAT videos is the perfect fit for military applications. It covers fundamental environment contexts (events), conditions (rather poor quality and weather condition impairments), and imagery (RGB and IR). Not including this dataset would mean not being representative of the UAV video ecosystem.

We decided to keep the original resolution of videos (720x480) not to include unrelated artifacts. Also, a lot of sensors embedded on UAVs present the same resolution. Lastly, it is important to keep in mind the reality of military operators. For instance, they can observe more than a percent of content per degree of visual angle during surveillance missions.

These precisions have been included in the text.

I would like to invite authors to ensure that the differences in number of point in calibration does not produce any substantial differences in error as this could reassure the reader that such change did not have any impact over the quality of data required.

This point is indeed interesting to note. However, we think it raises questions that have no influence on our results. We are indeed observing that a difference of 0.6 degree of visual angle in eye position makes almost no difference. The variation of accuracy that might be introduced by a change in calibration strategy is far less in magnitude and thus is considered insignificant here. Moreover, the calibration threshold guaranteeing the minimum accuracy level has not been changed.

The terminology « saliency maps » was used to designate convolved fixation maps. For clarity sake, I invite authors to change the terminology and refers to their computations as fixations maps instead of saliency maps as saliency maps refers to the maps extracted after computing the saliency from a saliency model.

Moreover, any usage of the word saliency that relates to gaze or fixations should be clarified. « Salience ground truth » can remain unchanged.

Terminology is indeed important. Thank you very much for bringing that on so that the data we report are precise and well understood. We relied on the following paper regarding the definition of saliency maps:

Kummerer, M.; Wallis, T.S.; Bethge, M. Saliency benchmarking made easy: Separating models, maps and metrics. Proceedings of the European Conference on Computer Vision (ECCV), 2018

Authors of this paper state that, before, saliency maps referred to “a map that predicts fixations”. Under this definition, we can use the term saliency maps in our context. However, the new definition attributed to saliency maps is the one provided in your review. Saliency maps should now be used exclusively for maps predicted by a saliency model. Even though we do not fully support this new definition, it is better to be consistent with the state of the art and not to add imprecision in the paper.

Calling our maps fixation density maps is not applicable in our case, as we do not compute the maps from fixations. We thus have selected the terminology gaze density maps.

In general, the Analyses section should be reworked in order to provide more details about the methodology used (see specific comments). I am questioning the usefulness of comparing left and right eye at the same time of Dominant and Non-dominant as the dominant eye tracking is the common method used in eye tracking studies. Moreover, evaluating both sides and dominance make Table 5 to be redundant.

Thus, in fine differences in average measurements for sides and dominance is the same and thus for the present purpose does not seems mandatory. Moreover this would help the ease to read analysis part.

Thank you for drawing our attention to the content and structure of the paper. Redundancy should indeed be avoided. Accordingly, we have removed the part on MAE results as we realized it provided rather limited insights. Doing so made some space to elaborate on statistical analyses. We also have restructured the paper. For instance, the new related work section encompasses all state-of-the-art information. We think this will lead to a more readable and clearer paper.

However, we believe that each of the six scenarios included in the analysis is compulsory to draw all our conclusions. For instance, including the four one eye scenarios enables us to conclude that dominance-based signals can be favored over the left and right eye signals. This conclusion could not be reached without including the four signals. Moreover, the fact that we do not have access to the real position of the eyes during the test defined our experiment design. We needed to have as many coherent and credible scenarios as possible to conduct a reliable cross-comparison. Solely comparing both eye and dominant eye signals would have significantly reduced the scope of the analysis.

When running ANOVA and multiple comparisons, authors should provide the entire statistical information for principal effects and interaction effects (F(X,X)=XX.XX,p<.05).

This is duly noted. We have included all the ANOVA results in a new table. See Table 5.

Afterwards multiple comparison is usually presented by comparing mean of the two compared group and indicating whether this effect is significant (p<.05) or not. All reference to p value below another threshold than .05 should be avoided (p<.10, p<.01, p<.001). Indeed such indication mislead to believe that p value indicates an effect size. I highly encourage authors to present their statistic in a more detailed way in order for the scientific community to evaluate the data analysis and the correctness of the statistics.

We can only agree with you on this point. However, in some cases, one may need to report different p-values, such as 0.01 (under 1% of chances to reject Ho while it should not). This is not applicable in our study so we changed back to the typical p-value set at 0.05.

I was really astonished to see that authors « discard » the non-significant metrics such as NSS. Indeed, NSS is now used by the MIT benchmark to sort model performance and is a very robust metrics (see Z Bylinskii, T Judd, A Oliva, A Torralba, F Durand What do different evaluation metrics tell us about saliency models? arXiv preprint arXiv:1604.03605, 2016; M Kammerer, T Wallis, M Bethge Information-theoretic model comparison unifies saliency metrics PNAS, 112(52), 16054-16059, 2015).

This is a really important comment. Thank you very much for pointing this out. Indeed, the word “discard” poorly expressed what was meant and introduced confusion. NSS must be included in benchmarks related to salience. Actually, we did include the metric. What we have not done though is to compute multi-comparison analyses on it. Indeed, it would be useless as no statistical difference has been detected with the ANOVA. However, we based our conclusions on the fact that for this metric, as well as AUC Judd and Borji, the scenarios are not different statistically. We tried to make these points clearer in the new version of the paper.

Hence, emphasis over significant results should be decrease in the light of other metrics that were found not to be significant and that are a ground truth in the field of saliency evaluation.

This is really true. See our answer above for more details.

Authors have to rework their results interpretations and the conclusion see specific comments. As their analysis do not support whether recording one eye is better than binocular recording. Indeed, comparison in the strength of correlation indicates only that the correlation between binocular and two eye is significantly higher than other correlation comparisons. Thus authors should make sure that their analysis allow to draw such conclusions.

We are glad that you bring this issue up. This means the first version of the paper did not correctly introduce our work. Multi-comparisons have been computed on CC, and also on SIM and two variants of IG. Relying only on CC is, we agree, reductive and only shows correlation, which does not necessarily represent causation for instance. We have rewritten the entire section describing multi-comparison tests and the analysis part, to address your comment. We have decided to include all the results of the metrics. Even though it introduces redundancies, it makes clear that saliency-specific metrics have been used during the analysis. It also shows the agreement between metrics.

Results should be rewrite to something like: « the correlation between binocular and both eyes recording is higher than both eye separately ». If authors want to claim that both eyes should be recorded then they must compare their condition using an ANOVA on a metric of performance of ocular movement on saliency and not on the correlation statistic.

We did our best to avoid it in the new version. However, let us note that we report the results of different metrics, including metrics of performance of ocular movement on saliency. Moreover, other metrics of saliency did not show significant differences and thus were not explored further with multi-comparison tests. In consequence, we found it is reasonable to use the vocabulary pointed out here.

One way to increase authors’ saliency ground truth would be to enrich it with a infinite human observer as described in Judd, Durand, and Torralba 2012. The evaluation of the differences in eye movements between their two conditions (free viewing and surveillance) would have been a plus.

This could have been performed using either grid segmentation/region of interest of the image or this one (Lao, J., Miellet, S., Pernet, C., Sokhn, N., & Caldara, R. (2017). iMap4: An open source toolbox for the statistical fixation mapping of eye movement data with linear mixed modeling. Behavior research methods, 49(2), 559-575).

It could have been interesting to evaluate EM according to the complexity of video and thus modulate ground truth with indices such as the ones used in the paper.

Thank you for the suggestions. We may apply these very interesting approaches in the future. However, we think that introducing more analysis in the paper may shade the contribution made with EyeTrackUAV2, in particular regarding the fading of center bias for UAV videos.

Specific comments:

L33 & P16 L469: Citation with XXX and colleagues [10] should be favored as "In [10]"

L45: Should be simplified by where point eyes or eyes fixates. Delete « more » in the second sentence.

P5 Table 1 is presented before the presentation of TI and SI indices. Hence, either these indices should be describe in table caption or the table should appear after indices explanation.

P6 Statistics (mean, std…) for SI and TI indices could have been provided in table 2

L305-308: This paragraph is not clear to me. Either clarify or delete it.

L361-362: How did you do that ? Please develop the methodology used here.

L437-444: The formulation while presenting the correlation analysis should be rewrite. Indeed,« The worst scenario, significantly in Task » mislead that authors’ correlation analysis could provide evaluation the best scenario whereas it only indicate whether two metrics were correlated.

Thank you a lot for all these precisions. We have made sure to address every single one in the new version of the paper.

Again, you have all our gratitude for having pointed out the discrepancies in the writing and the potential points to correct or elaborate. We have done our best to address your comments.

As a side note, we here provide the credentials for accessing the private FTP so that you can fully review our work. Here is the information you need:

The correct address to connect to the private FTP is: ftp://[email protected]/EyeTrackUAV2/

The user id is:            dissocie

And the password:     KOo1NVlg

Round 2

Reviewer 1 Report

Thank you for the revision. The authors have corrected the article based on the comments given in the first review. 

Author Response

We would like to thank again the reviewer for all his comments. We believe he pointed out unclear information that had been provided in the paper. Now, we are positive the new version is complete, understandable and transparent.

Reviewer 2 Report

I think that authors provide a substantial improvement of the manuscript. They clarified the whole paper and most of my point were well addressed. I would like to thank the authors for their hard work.

While on my opinion the terminology saliency map should still be avoided. Authors clarified this point and provide their own definition. Moreover, the evaluation of the maps is clearer now.

I still have several comments that are to adress:

While, complexity measurements were added in table 3, the point raised was to add this measure for all dataset in order for the reader to be able to compare the datasets used. I apologize if my comment was not clear enough.

In the first version of the manuscript, authors mentioned a differences in the calibration procedure. However, in this corrected version this disappear. Hence, for transparency authors must include and specify how they performed calibration as in the first version of this manuscript.

I would like to thanks the authors for the significant improvement of the analysis description and report. However, this part still suffers from the fact that analysis do not support the claims of authors. Indeed, despite claims are less strong than before they do not reflect results obtained.

If authors would like to encourage or provide recommendation about eye movements recording. To perform an analysis performing as said in the analysis section L399-401: « Scenarios highly similar to all others are the ones that will make fewer errors ». Indeed, from now one this description do not reflect the analysis performed. The proposition at L399-401 is statistically sound and could be use to make the claims authors wanted.

One way to perform this analysis would be to get all CCs according to variables compares. One columns will represent the scenario used (B, BE, L, R, D, ND) and another one encompassing the 5 CCs, e.g.:
C1           C2
Binocular .94
Binocular .95

Binocular .98
Both Eyes .98
Both Eyes .96
etc.

Then, a regular ANOVA can be done on this table. Of course this can be applied to the all the measurements.

On the contrary, I would like authors to stick to their data and analysis and completely delete any suggestions and recommendations about the « best » scenario.

I would really like to stress out the importance of this analysis in order for the authors to have sound claims.

Author Response

First, we would like to thank again the second reviewer for all his comments. We believe he pointed out all the unclear information that had been provided in the paper. Thanks to these conditions, the new version is clearer, understandable and transparent.

I think that authors provide a substantial improvement of the manuscript. They clarified the whole paper and most of my point were well addressed. I would like to thank the authors for their hard work.

Thank you very much. We indeed put a lot of effort to address comments to improve the paper quality and clarity.

While on my opinion the terminology saliency map should still be avoided. Authors clarified this point and provide their own definition. Moreover, the evaluation of the maps is clearer now.

That is unfortunate. We were sure having changed all terminologies accordingly to your first comment. We went through the entire paper again and verified when "saliency maps" have been used. Depending on the context, we changed it to gaze density maps or human saliency maps to precise our thoughts and ensure the correctness of the paper.

I still have several comments that are to adress:

While, complexity measurements were added in table 3, the point raised was to add this measure for all dataset in order for the reader to be able to compare the datasets used. I apologize if my comment was not clear enough.

This is a very good point. Thank you for noting it. We did not get this point accurately the first time. Sorry for the inconvenience. It is corrected in the new version. For the sake of completness, we have included the information in Table 1.

In the first version of the manuscript, authors mentioned a differences in the calibration procedure. However, in this corrected version this disappear. Hence, for transparency authors must include and specify how they performed calibration as in the first version of this manuscript.

Transparency is highly important to us. We changed back this paragraph to the first version.

I would like to thanks the authors for the significant improvement of the analysis description and report. However, this part still suffers from the fact that analysis do not support the claims of authors. Indeed, despite claims are less strong than before they do not reflect results obtained.

If authors would like to encourage or provide recommendation about eye movements recording. To perform an analysis performing as said in the analysis section L399-401: « Scenarios highly similar to all others are the ones that will make fewer errors ». Indeed, from now one this description do not reflect the analysis performed. The proposition at L399-401 is statistically sound and could be use to make the claims authors wanted.
One way to perform this analysis would be to get all CCs according to variables compares. One columns will represent the scenario used (B, BE, L, R, D, ND) and another one encompassing the 5 CCs, e.g.:
C1           C2
Binocular .94
Binocular .95

Binocular .98
Both Eyes .98
Both Eyes .96
etc.

Then, a regular ANOVA can be done on this table. Of course this can be applied to the all the measurements.

On the contrary, I would like authors to stick to their data and analysis and completely delete any suggestions and recommendations about the « best » scenario.
I would really like to stress out the importance of this analysis in order for the authors to have sound claims.

This comment has triggered our attention. It has been clear Reviewer does not find our conclusions sound. This is a serious reaction to the study. Especially because we have made sure that the analysis is proper and reliable.

At first, we had trouble understanding this comment. In our opinion, we have done what Reviewer proposed. After carefully reading your review, it seems there has been a confusion.

If we understood Reviewer’s statement well, Reviewer are proposing to run a statistical analysis on all comparisons of scenarios. We actually did it. What was named multi-comparison is Tukey’s multiple comparison post-hoc test (with alpha = 0.05). It is used following an ANOVA to identify which comparisons have been found statistically different from others.

The wording may have been unfortunate. Despite this fact, to our mind and based on the processing used, all conclusions and claims are based on statistical tests and are technically sound. Accordingly, we did not change the conclusions. However, to make the point clearer, we have added the description of the entire analysis of the multiple comparison test.

Briefly, this is what ANOVA and Tukey’s test tell us (see Figure 6 and Table 6)
(I) NSS, AUC J, AUC B ->

No statistical difference observed
(II) CC, SIM IGs ->

There are 3 statistically different classes for all metrics, for all attention conditions :
    (1)
Comparisons with B et BE, together with one eye scenarios representing the same eyedness (e.g. D vs L and nD vs R for FV, D vs R and nD vs L for Task)
    (2) One eye scenarios including different eyedness information (i.e. D vs R and nD vs L for FV, D vs L and nD vs R for Task)
    (3) Inverse one eye scenarios (i.e. L vs R and D vs nD)
(III)SIM et IG SM2-Fix2-SM1 en FV et Task, et CC Task ->

BE vs B statistically different than all other scenarios. This comparison achieves the best group mean.

(IV) CC, SIM, IGs Task et SIM FV ->

L vs R statistically different from D vs nD

II-1 and III pledge for using both eyes information.
II-2 and II-3 tell us eyedness impacts data. IV show statistically that, at least for Task, dominance-based one-eye signals are more representative than L and R.
II-1 is a stronger finding than IV regarding the number of metrics on which we base the analysis and if the conclusion extracted stands for all attention conditions.
All findings must be tempered due to finding I.
We can conclude that both-eye signals represent better gaze information. Dominance-based one-eye signals may be used if both eye information is not available. Again, these are not strong recommendations as three metrics could not differentiate scenarios through the ANOVA.

Finally, we value the preciseness and all your efforts put in Reviewer comments. That enables us to cut down misinterpretations or unclear/unprecise information. Thank you for this.